# Effects of experiencing the COVID-19 pandemic on optimistically biased belief updating

Iraj Khalid[1], Orphee Morlaas[1], Hugo Bottemanne[1,2], Lisa Thonon[1], Thomas Da Costa[1], Philippe Fossati[1,2], Liane Schmidt[1]*

[1]Control-interception-attention team, Paris Brain Institute (ICM), UMR 7225, U1127, Institut National de la Santé et de la Recherche Médicale/Centre National de la Recherche Scientifique/Sorbonne Universités, Hôpital Pitié-Salpêtrière, Paris, France; [2]Département de Psychiatrie Adulte, Hôpital Pitié-Salpêtrière, Assistance Publique Hôpitaux de Paris (APHP), Paris, France

## eLife Assessment

This **important** study addresses the question of how large-scale events such as the COVID-19 pandemic can change people's beliefs and their updates. Using a well-validated task, the authors find that belief updating becomes less optimistically biased during COVID-19 compared to before it. In this revision, due to the addition of more model-based analyses and power calculations, they have generated **convincing** evidence for their primary claim that the pandemic significantly impacted people's belief updating away from optimistic belief updating. As with many manipulations outside the experimenters' control, it remains unclear which psychological factor impacted by the pandemic drives the group differences, and sample sizes are, by necessity, on the smaller side as data cannot readily be acquired. However, the authors are commended for doing power analyses, showing their sensitivity, and recognizing the limitations of their study.

**\*For correspondence:**
liane.schmidt@icm-institute.org

**Abstract** Optimistically biased belief updating is essential for mental health and resilience in adversity. Here, we asked how experiencing the COVID-19 pandemic affected optimism biases in updating beliefs about the future. One hundred and twenty-three participants estimated the risk of experiencing adverse future life events in the face of belief-disconfirming evidence either outside the pandemic (n=58) or during the pandemic (n=65). While belief updating was optimistically biased and Reinforcement-learning-like outside the pandemic, the bias faded, and belief updating became more rational Bayesian-like during the pandemic. This malleability of anticipating the future during the COVID-19 pandemic was further underpinned by a lower integration of positive belief-disconfirming information, fewer but stronger negative estimations, and more confidence in base rates. The findings offer a window into the putative cognitive mechanisms of belief updating during the COVID-19 pandemic, driven more by quantifying the uncertainty of the future than by the motivational salience of optimistic outlooks.

## Introduction

Anticipating the future is an essential part of human thinking. These beliefs guide how we understand the world; updating them is vital for learning to make better predictions and to generalize across contexts. Interestingly, belief updating tends to be optimistically biased (*Weinstein, 1980*). Even when confronted with negative information, humans often downplay its importance to maintain

**eLife digest** Anticipating the future is an essential part of human thinking and helps us guide how we understand the world. We continuously update this information to make better predictions and generalizations.

When faced with new information, we tend to downplay bad news and emphasize good news. This bias towards optimism supports mental health and helps people stay motivated and resilient when facing difficulties.

However, in hard times, and/or when people feel threatened or depressed, they may lose their ability to focus on good news. Instead, they may begin to emphasize possible negative outcomes. For example, the COVID-19 pandemic created a long-lasting period of challenges for many people. Every-day activities, like going to work or food shopping, became risky. Uncertainty about the economy and health impacts further weighed on people's minds.

Khalid et al. set out to understand if extreme and long-lasting challenges affect the optimism bias in belief updating. The experiments compared how 93 adults estimated their chances of adverse experiences like a job loss or illness during the pandemic in France, with the estimations of 60 individuals before or after the pandemic. More specifically, 30 people were recruited in France in October 2019 before the pandemic, and a further 34 and 31, respectively, were recruited in March to April 2020 and May 2021, and in addition 28 people from the pre-pandemic group were consulted again in March to April 2020. Another 30 people were recruited after the pandemic in June 2022.

The experiments measured how much people's predictions changed after learning real statistics about the likelihood of adverse future life events. Before and after the pandemic, people showed a strong optimistic bias when considering these statistics. They gave more weight to good news than to bad news. But this bias disappeared during the pandemic, and people weighed positive and negative information more evenly.

The experiments also showed that before and after the pandemic, people updated their beliefs in a way that resembled trial-and-error learning. In contrast, during the pandemic, their updates were more cautious and evidence-based, like weighing all the facts before making a judgment.

Khalid et al. shed light on how a large-scale crisis can affect our perception of the future. In the long term, this could help mental health professionals better understand how people cope with uncertainty and guide efforts to strengthen resilience in difficult times. More research is needed to explore how changes in belief updating relate to mental health and what it means to adopt a healthy level of optimism to recover after significant life challenges.

---

an optimistic view of the future. The underlying mechanism of optimistically biased belief updating involves an asymmetry in learning from positive and negative belief-disconfirming information (*Sharot et al., 2011*; *Sharot and Garrett, 2016*; *Kuzmanovic et al., 2018*), which can unfold in two ways, following Reinforcement learning (RL) or Bayes rule (*Kuzmanovic and Rigoux, 2017*).

Conceptually, Reinforcement learning (RL) and Bayesian models of belief updating are complementary but make different assumptions about the hidden process humans may use to adjust their beliefs when faced with information that contradicts them. The RL models assume that belief updating is proportional to the estimation error. The key idea of the estimation error expresses the difference between how much someone believes, for example, they will experience a future life event and the actual prevalence of the event in the general population. This difference can be positive or negative. A scaling and an asymmetry parameter quantify the propensity to consider the estimation error magnitude and its valence, respectively. These two free parameters form the learning rate, which indicates how fast and biased participants update their beliefs.

In contrast, Bayesian models assume that following Bayes' rule the posterior, updated belief is a new hypothesis, formed by pondering prior knowledge with new evidence. The prior knowledge consists in information about the prevalence of life events in the general population. The new evidence comprises various alternative hypotheses. It examines how likely a specific event is to occur or not occur for oneself, compared to the likelihood that it will happen or not happen to others. This probabilistic adjustment of beliefs about future life events can be considered as an approximation of a participant's confidence in the future. The two free parameters of the Bayesian belief updating model

scale how much the initial belief deviates from the updated, posterior belief (i.e. scaling parameter) and the propensity to consider the valence of this deviance (i.e. asymmetry parameter).

Although RL-like and Bayesian updating models make different assumptions about the updating strategy, they are complementary and powerful formalizations of human reasoning. Both models provide insight into hidden, latent variables of the updating process. Most notably, the learning rate and its components, the scaling and asymmetry parameters, can vary between individuals and contexts and, through this variance, offer possible explanations for the idiosyncrasy in belief-updating behavior and its cognitive biases.

The COVID-19 pandemic represented a chronic adverse life context, drastically altering individual and social realities. The existing literature documenting the impact of the COVID-19 pandemic and the associated changes in everyday life on mental health has shown that stress, anxiety, and depression have increased during the pandemic (*Rajkumar, 2020*; *World Health Organization, 2022*). Moreover, previous work has demonstrated that belief updating during reversal learning became more erratic and was linked to a rise in paranoia during the COVID-19 pandemic across the US (*Suthaharan et al., 2021*).

However, it is unknown how optimism biases in belief updating about the future and their underlying putative mechanisms changed during the experience of such an unprecedented, adverse life event. Our hypothesis was twofold. We argued that maintaining optimistically biased belief updating under lasting, adverse life conditions is adaptive. The optimism biases are beneficial in exploratory behavior, reduce stress, and improve mental and physical health and well-being (*Taylor and Brown, 1988*; *Scheier et al., 1994*; *Scheier et al., 2001*; *Boehm and Lyubomirsky, 2008*; *Carver et al., 2010*; *Carver and Scheier, 2014*; *Boehm et al., 2018*). These benefits promote resilience, which is especially important for fitness and survival during a pandemic (*Berger-Tal and Avgar, 2012*; *McKay and Dennett, 2009*). On the contrary, optimism biases can lead to suboptimal decision-making. Contextual factors such as acute stress, perceived threat, and depression have been shown to reduce or even reverse optimistically biased belief updating (*Strunk et al., 2006*; *Garrett et al., 2014*; *Korn et al., 2014*; *Garrett et al., 2018*; *Kuper-Smith et al., 2020*; *Czekalla et al., 2021*; *Bottemanne et al., 2022*). These findings suggest that optimistically biased belief updating should be weaker when experiencing a pandemic.

We leveraged a belief-updating dataset from 123 participants tested between 2019 and 2022 to rule between these alternative hypotheses. Among them, 58 participants were tested outside the context of the COVID-19 pandemic, either in October 2019, 3 months before the outbreak in France (n=30) or 2 years after in June 2022 (n=28), after the lift of the sanitary state of emergency. Their belief updating behavior was compared to 65 participants tested during the sanitary state of emergency due to the COVID-19 outbreak in France. This was either during the first very strict lockdown of social and economic life (e.g. schools closed, stay-at-home orders, shops, and museums closed) from March to April 2020 (n=34) or 1 year later in May 2021 (n=31), when lockdown was less strict (e.g. schools open, museums and shops closed, part-time curfew), but the COVID-19 pandemic was still unfolding. Belief updating was measured by a behavioral task that asked participants to estimate their risk of experiencing adverse future life events before and after receiving information about these events' actual base rates. Observed belief updating behavior was fitted to an RL-like and a Bayesian updating model to gain insight into potential underlying strategies of belief updating. The learning rates were compared across groups for insight into how experiencing the COVID-19 pandemic changed beliefs about the future and their updating in the face of belief-disconfirming evidence.

## Results
### Effects of experiencing the COVID-19 pandemic on optimistically biased belief updating

A linear mixed effects (LME) model was fitted to belief updates to test whether belief updating was less or more biased during the COVID-19 pandemic. The model found a significant interaction estimation error valence by context (ß=−5.54, SE = 1.69, t(232) = −3.28, p=0.001, 95% CI [-8.87 to −2.21]; *Appendix 7—table 2*), which holds when further controlling for the distance variable (*Appendix 7—table 3*). The power of this effect was 75% leaving a 25% risk for type II errors. As shown in *Figure 1a and b*, optimistically biased belief updating disappeared during the COVID-19 pandemic compared

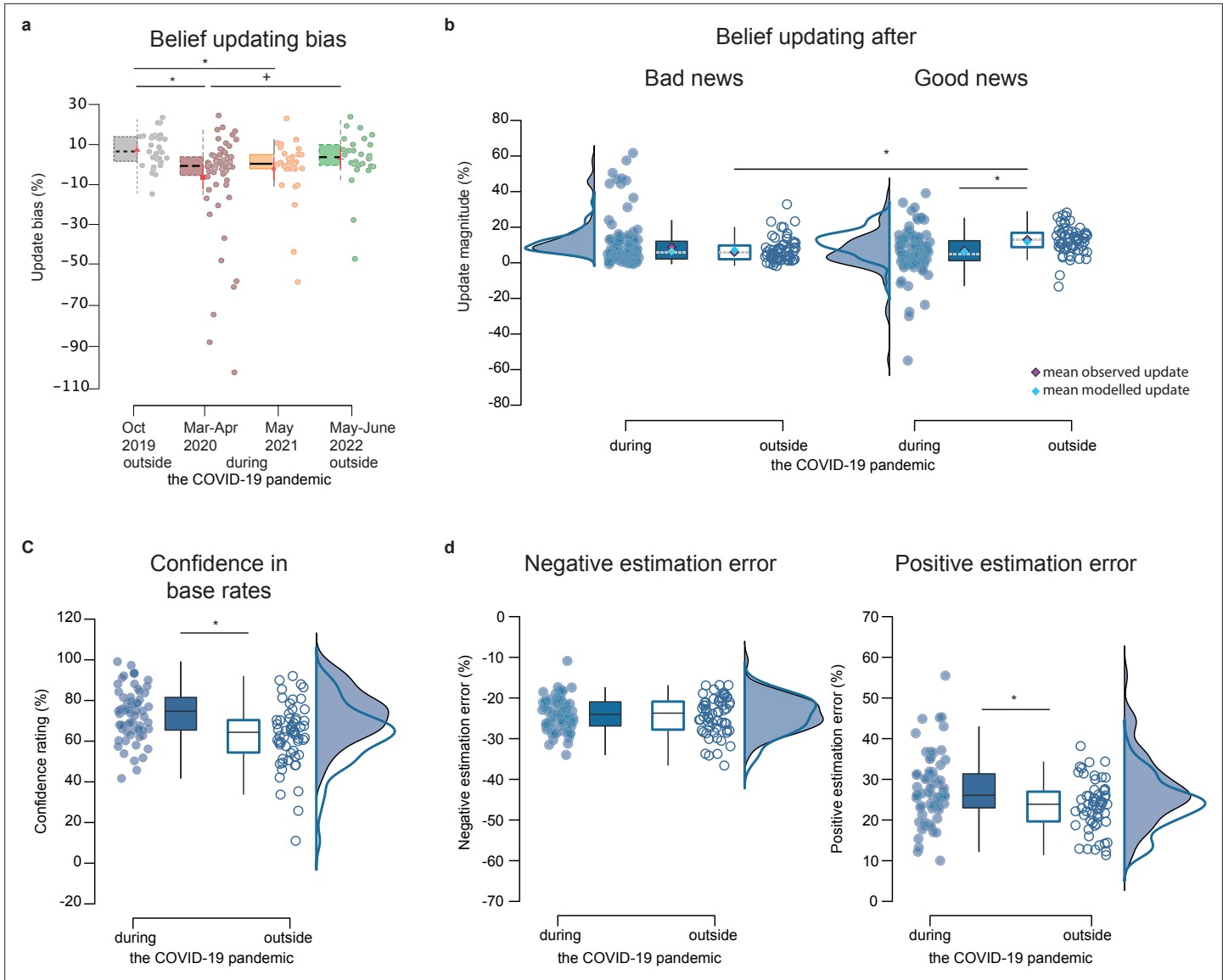

**Figure 1.** Behavioral results. (**a**) *Boxplots display the belief-updating bias* (i.e. the difference between the belief update for good news and belief update for bad news) in each of the four participant groups, tested before the pandemic in October 2019 (n=30), during the first lockdown from March to April 2020 (n=34), with less restrictive measures in May 2021 (n=31), and at the end of the pandemic in June 2022 (n=28). (**b**) *Belief updating for good and bad news* during (n=65) and outside the pandemic (n=58). (**c**) *Confidence ratings*, and (**d**) *estimation errors for bad and good news during and outside the pandemic*. Boxplots in all panels display 95% confidence intervals, with boxes indicating the interquartile range from Q1 25th to Q3 75th percentile. The horizontal black lines indicate medians, and whiskers range from minimum to maximum values and span 1.5 times the interquartile range. The dots correspond to individual participants. The squares in the boxplots in (**b**) correspond to mean observed updates (purple) and mean modelled updates (blue; averaged across 1000 estimations) from the best-fitting models in each context, which were the optimistically biased RL-like model of belief updating outside and the rational Bayesian model of belief updating during the Covid-19 pandemic. The source data file provides exact values. *<0.05 two-sampled, two-tailed t-tests, * p<0.05 two-sampled, one-tailed t-tests.

The online version of this article includes the following source data and figure supplement(s) for figure 1:

**Source data 1.** Average values across trials for each participant on behavioral outcome measures.

**Figure supplement 1.** Belief updating within the same group of participants tested before and during the COVID-19 pandemic (n=28).

**Figure supplement 1—source data 1.** Average belief update across trials for each participant tested both before and during the pandemic (within-subjects).

**Figure supplement 2.** Optimism bias in initial beliefs about adverse future life events.

**Figure supplement 2—source data 1.** Average first estimate across trials for each participant.

to participants tested outside the pandemic. More specifically, it was decreased among participants tested during the initial COVID-19-related strict lockdown in March and April 2020 (EE valence by context 1: ß=–7.39, SE = 2.29, t(228) = –3.21, p=0.002, 95% CI [-11.91 to –2.86]; *Appendix 7—table 4*), as well as in May 2021 (EE valence by context 2: ß=–5.59, SE = 2.36, t(228) = –2.37, p=0.02, 95% CI [-10.24 to –0.93]; *Appendix 7—table 4*), compared to those tested before the outbreak in October 2019, respectively. The bias re-emerged among participants tested one year later at the time of the lift of the sanitary state of emergency in June 2022, returning to levels akin to those observed before the pandemic in October 2019 (EE valence by context 3: ß=–2.11, SE = 2.46, t(228) = –0.86, p=0.39, 95% CI [-6.95 to 2.73]; *Figure 1a*, *Appendix 7—table 4*). The effect of the COVID-19 pandemic on belief updating was driven by a significant decrease in belief updating following good news during the pandemic compared to participants tested outside the pandemic (t(121) = 2.66, p=0.009, Cohen's d=0.48, two-sampled, two-tailed t-test, *Figure 1b*). No contextual group difference was observed for belief updating following bad news (t(121) = –1.77, p=0.08, Cohen's d=–0.32, two-sampled, two-tailed t-test, *Figure 1b*). This effect could be reproduced when fitting an analogous LME to belief updates observed in the group of participants (n=28) who were tested both before and during the pandemic (EE valence by context interaction: ß=–7.66, SE = 1.49, t(103) = –5.13, p=1.35e-06, 95% CI [-10.62 to -4.70]; *Figure 1—figure supplement 1*, *Appendix 7—table 5*, Appendix 1). Moreover, previous studies of optimistically biased belief updating calculated the estimation error (EE) on the difference between the estimate for someone else (eBR) and the base rate (BR), following: EE = eBR - BR (*Kuzmanovic et al., 2018*; *Kuzmanovic and Rigoux, 2017*; *Garrett and Sharot, 2014*; *Kuzmanovic et al., 2015*). When categorizing trials as good or bad news based on this alternative EE calculation, the context-by-EE valence interaction remained significant (*Appendix 7—table 6*). Note that all effects were controlled for participants' age, years of higher education, gender, confidence in the base rates, belief updating task design, and estimation error magnitude.

## Effects of experiencing the COVID-19 pandemic on belief updating variables

As shown in *Figure 1c*, experiencing the COVID-19 pandemic influenced participants' confidence in the base rates, with significantly lower confidence ratings observed among those tested outside the pandemic compared to those tested during it (ß=14.11, SE = 4.52, t(233) = 3.12, p=0.002, 95% CI [5.19 to 23.02]; *Appendix 7—table 7*). Moreover, a significant interaction of EE valence by context (ß=2.19, SE = 0.67, t(233) = 3.28, p=0.001, 95% CI [0.88 to 3.51]; *Appendix 7—table 8*) was found for absolute estimation error magnitude. This finding indicated that participants tested during the pandemic overestimated their risk of experiencing adverse future life events relative to base rates more largely than participants tested outside the pandemic (t(121) = –3.01, p=0.003, Cohen's d=–0.54, *Figure 1d*). On the contrary, the two groups did not differ significantly in the magnitude of negative estimation errors (i.e. initial underestimations relative to base rates; t(121) = –0.49, p=0.63, Cohen's d=–0.09, two-sampled, two-tailed t-tests; *Figure 1d*). This finding contrasts with the observed difference in how often they made positive estimation errors (i.e. the number of good news trials). Participants tested during the pandemic overestimated less frequently than participants tested outside the pandemic (t(121) = 2.40, p=0.02, Cohen's d=0.43, two-sampled, two-tailed t-test). No significant difference between groups was found for the frequency of underestimations (i.e. reflected by the number of bad news trials; t(121) = –1.85, p=0.07, Cohen's d=–0.33, two-sampled, two-tailed t-test). These results indicated that participants held fewer but stronger negative future outlooks during the pandemic compared to those tested outside the pandemic.

## Effects of experiencing the COVID-19 pandemic on putative mechanisms of belief updating

We then sought to identify which putative strategy participants used to update their beliefs about the future during and outside the pandemic. To answer this question, we used computational modeling and model comparisons to rule between 12 alternative models. This approach revealed that belief updating outside the pandemic was more RL-like and optimistic (pxp = 1, Ef = 0.77), while during the pandemic, it was best explained by a rational Bayesian updating model (pxp = 0.90, Ef = 0.43; *Figure 2a–c*). Similar findings were obtained when conducting model comparisons in the participants tested both before and during the lockdown (n=28; Appendix 1 , *Figure 2—figure supplement 1*).

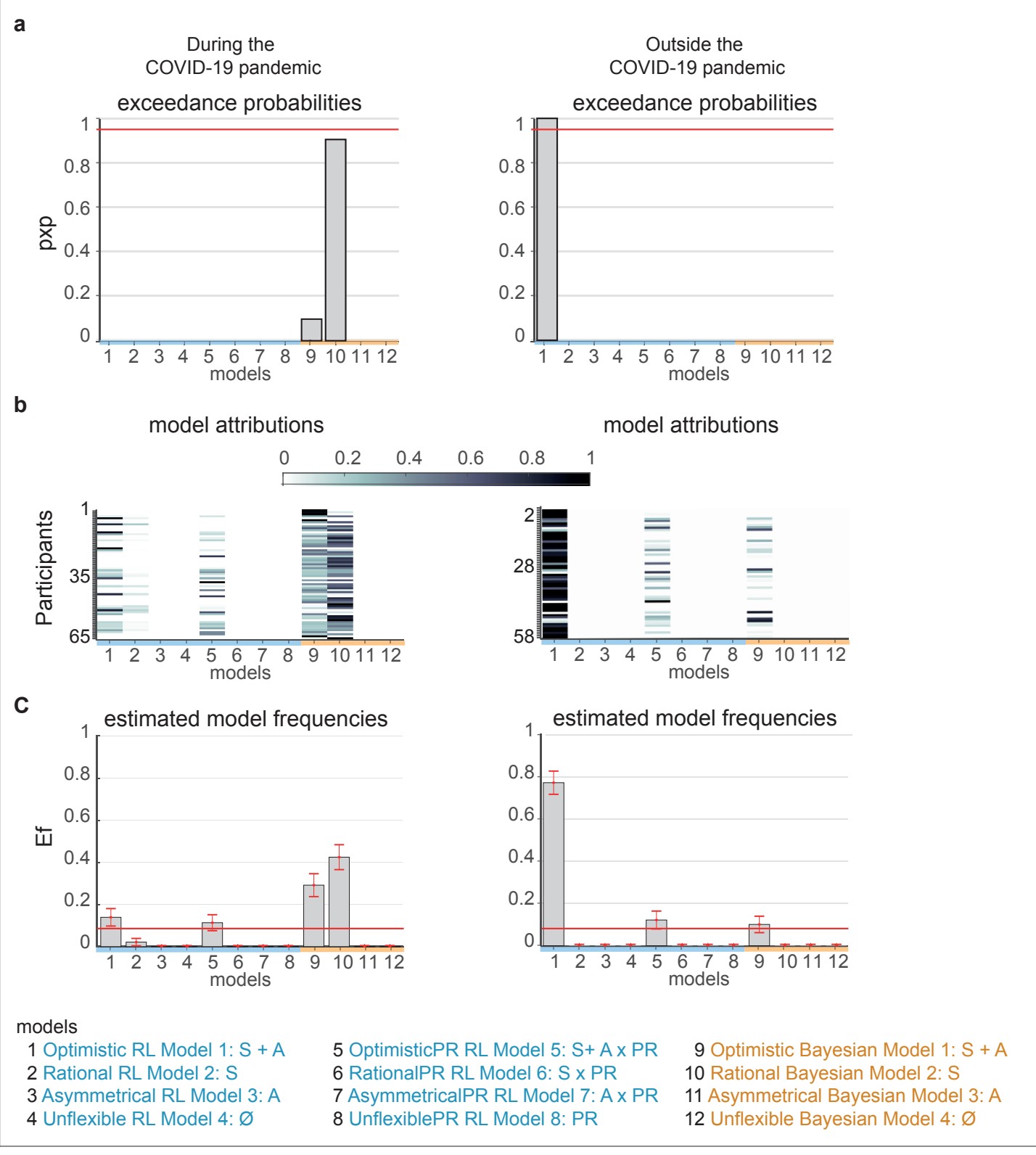

**Figure 2.** Computational model comparisons. Twelve alternative models from RL-like (blue) and Bayesian (orange) updating model families were fitted to observed belief updates for participants tested during the COVID-19 pandemic (left panel columns) and outside the pandemic (right column panels). (**a**) *Protected exceedance probabilities for each of the 12 alternative models*, which is the probability that the model predominates in the population above and beyond chance. (**b**) *Posterior model attributions*. Colored cells display the probability that individual participants (y-axis) will be best explained by a model version (x-axis). (**c**) *Estimated model frequencies* correspond to how many participants are expected to be best described by a

*Figure 2 continued on next page*

*Figure 2 continued*

model version, with error bars corresponding to standard deviations. The red line indicates the null hypothesis that all model versions are equally likely in the cohort (chance level). Labels on the x-axis of the histogram and bar graphs indicate the model versions with non-silenced parameters (S – scaling, A – asymmetry) and PR – personal relevance of events. The source data file provides exact values.

The online version of this article includes the following source data and figure supplement(s) for figure 2:

**Source data 1.** Model comparison metrics.

**Figure supplement 1.** Estimated model frequencies for participants tested both before and during the COVID-19 pandemic.

**Figure supplement 1—source data 1.** Model comparison metrics for the within-subjects analysis.

**Figure supplement 2.** Model recovery confusion matrix.

**Figure supplement 2—source data 1.** Estimated model frequencies from the model recovery analysis.

**Figure supplement 3.** Observed and modelled belief updating for the whole participant sample (n=123).

**Figure supplement 3—source data 1.** Average observed and modelled belief update across trials for each participant.

## Effects of experiencing the COVID-19 pandemic on hidden, latent variables of belief updating

Next, we compared the effects of experiencing the COVID-19 pandemic on the learning rates and its components. To show that this adverse context effect was indeed mediated by alterations in asymmetrical learning, we compared the scaling and asymmetry parameters obtained from the overall best-fitting model across the whole dataset of n=123 participants. This was Model 1 – the optimistically biased RL-like model of belief updating (pxp = 0.99, Ef = 0.40; Appendix 4, *Figure 2—figure supplement 3*).

A linear mixed effects model (LME), analogous to the LME fitted to observed belief updates, was fitted to the learning rates and detected a main effect of EE valence (ß=0.09, SE = 0.01, t(236) = 7.14, p=1.18e-11, 95% CI [0.06 to 0.11]; *Appendix 7—table 9*), and a significant interaction EE valence by context (ß=–0.03, SE = 0.02, t(236) = –2.11, p=0.04, 95% CI [-0.07 to –0.002]; *Figure 3a*, *Appendix 7—table 9*). A main effect of EE valence (ß=0.08, SE = 0.02, t(105) = 3.22, p=0.002, 95% CI [0.03 to 0.12]; *Appendix 7—table 10*) and context (ß=–0.10, SE = 0.03, t(105) = –3.10, p=0.003, 95% CI [-0.17 to –0.04]; *Appendix 7—table 10*) on learning rates was detected when comparing the participants, who were tested both before and during the pandemic. As shown in *Figure 3a*, all participants' learning rates were lower in response to bad news than to good news. Still, the difference between good and bad news learning rates was significantly reduced for participants tested during the pandemic. In line with the observed belief updating after good and bad news, the effect of context on the learning rates was driven by a decrease in the learning rates from positive estimation errors in participants tested during the pandemic compared to participants tested outside the pandemic (t(121) = 2.17, p=0.03, Cohen's d=0.39, two-sampled, two-tailed t-test). Both groups did not differ in their learning rates from negative estimation errors (t(121) = 0.87, p=0.39, Cohen's d=0.16, two-sampled, two-tailed t-test).

Parameter recovery was successful for the scaling (r=0.92, p<0.001) and asymmetry (r=0.82, p<0.001) components of the learning rates (*Figure 3b*), which indicated that the model gave identifiable values for these parameters (parameter recovery was also conducted on each group and each model family separately, results are reported in Appendix 4 and *Figure 3—figure supplement 1*). We, therefore, were able to explore potential group differences in the learning rate components in more detail. Linear mixed effects modeling found a main effect of context for the asymmetry component (ß=–0.04, SE = 0.02, t(117) = –2.32, p=0.02, 95% CI [-0.07 to –0.01]; *Figure 3c*, *Appendix 7—table 11*), but not for the scaling component (ß=–0.07, SE = 0.05, t(117) = –1.54, p=0.13, 95% CI [-0.16 to 0.02]; *Figure 3c*, *Appendix 7—table 12*). The average asymmetry of learning rates was positive in both groups but significantly smaller in participants tested during the pandemic than those tested outside (t(121) = 2.00, p=0.048, Cohen's d=0.36, two-sampled, two-tailed t-test, *Figure 3c*). This result indicated that participants considered positive estimation errors more than negative ones but less when experiencing the COVID-19 pandemic. Similar results were found in the within-subject group (n=28), with a significant main effect of context on asymmetry (ß=–0.06, SE = 0.02, t(51) = –3.72, p=0.001, 95% CI [-0.09 to –0.03]; *Appendix 7—table 13*), but not on scaling (ß=–0.10, SE = 0.05, t(51) = –1.96, p=0.06, 95% CI [-0.21 to 0.003]; *Appendix 7—table 14*).

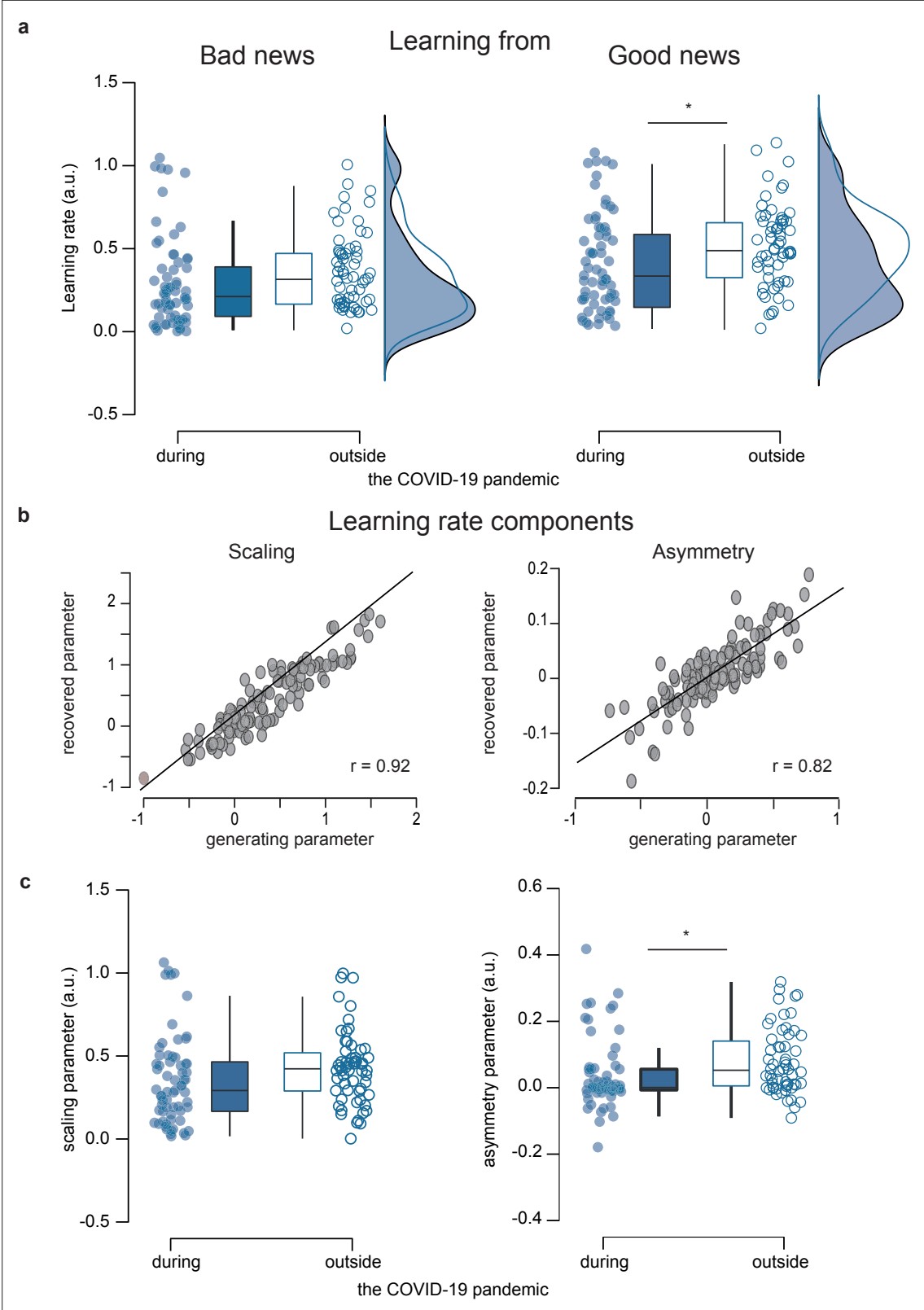

**Figure 3.** Parameter comparisons between participants tested during (n=65) and outside (n=58) the COVID-19 pandemic. (**a**) Learning rates. Boxplots display 95% confidence intervals for learning rates from the RL-like updating model that assumed updating is proportional to the estimation error with an asymmetry and a scaling learning rate component. (**b**) *Parameter recovery for learning rate components of the overall best fitting Model 1 (n=123).* Pearson's correlation between generating and recovered parameters for scaling (left panel) and asymmetry (right panel) learning rate component. r

*Figure 3 continued on next page*

*Figure 3 continued*

–Pearson's correlation coefficient against zero. Source data and exact p-values are provided as a Source Data file. (**c**) *Group comparisons for scaling and asymmetry components*. Boxplots display 95% confidence intervals for the learning rate's scaling (left panel) and the asymmetry (right panel) component. Boxes in all boxplots correspond to the interquartile range from Q1 (25th percentile) to Q3 (75th percentile). The horizontal black lines indicate medians, and whiskers range from minimum to maximum values and span 1.5 times the interquartile range. The dots correspond to individual participants. *p<0.05. p-values were obtained with two-sampled, two-tailed t-tests between groups, and exact values are provided in the source data file.

The online version of this article includes the following source data and figure supplement(s) for figure 3:

**Source data 1.** Computational model parameters per participant.

**Figure supplement 1.** Parameter recovery for the wining model family according to context.

**Figure supplement 1—source data 1.** Pearson correlation coefficients per participant for parameter recovery.

## Discussion

This study investigated how experiencing the COVID-19 pandemic impacted the optimism biases in updating beliefs about the future. Belief updating was optimistically biased before the COVID-19 outbreak, faded during the COVID-19 pandemic, and reemerged after the pandemic. The lack of optimistically biased belief updating during the pandemic was related to three effects: (1) a decreased sensitivity to positive belief-disconfirming information, (2) fewer but stronger negative beliefs about the future, and (3) more confidence in base rates. Computational modeling showed that belief updating during the pandemic was best described by a rational Bayesian model. In contrast, an optimistic RL-like model best-approximated belief updating outside the pandemic. Both models showed that the attenuated optimistically biased belief updating during the pandemic was not due to a learning deficit. The groups were similar in how much they integrated overall evidence in favor of or against initial beliefs. On the contrary, it was explained by a diminished learning asymmetry in considering positive belief-disconfirming evidence that paralleled the observed belief-updating behavior.

The finding that optimistically biased belief updating faded during the pandemic favors the hypothesis that experiencing an adverse life event such as a pandemic weakens optimistic outlooks. It further aligns with the body of research that has explored the malleability of the optimistically biased belief updating and information integration under acute threat and stress and mood disorders such as depression (*Strunk et al., 2006*; *Garrett et al., 2014*; *Korn et al., 2014*; *Garrett et al., 2018*; *Kuper-Smith et al., 2020*; *Czekalla et al., 2021*; *Bottemanne et al., 2022*; *Globig et al., 2021*). While our findings align with these previous findings, we also observed a difference. Notably, our sample tested during the pandemic considered positive, favorable information less while showing no change in negative, unfavorable information consideration. Differences in populations and task designs might explain these odds. However, it could also be specific to experiencing the COVID-19 pandemic, which involved an immediate, unpredictable, and global health threat with high uncertainty about its outcome and significant psychological repercussions (*Rajkumar, 2020*; *Brooks et al., 2020*).

Mental health assessments during the COVID-19 pandemic indicated that anxiety, stress, paranoia, and depression levels were more prevalent in the population (*World Health Organization, 2022*; *Suthaharan et al., 2021*). The rapid spread of SARS-CoV-2 and the emergence of COVID-19 cases worldwide constitute a challenging situation that, in 5 months, has shifted from an elusive and distant threat to an immediate and drastic health and economic crisis. All citizens were confronted daily with alarming figures such as infection rates or mortality, and rather mundane everyday activities, from grocery shopping to jogging, became stressful and threatening situations during which one could catch a potentially fatal infection. Moreover, for many, the COVID-19-related lockdowns of social and economic life implied a physical cut-off from friends and relatives, and life plans, routines, and activities were severely disrupted. It also implied a substantial economic risk. Previous research has shown that economic uncertainties, particularly during marked economic inequality and epidemics, can contribute to belief-updating fallacies reflected by the rise of conspiracy theories (*Leonard and Philippe, 2021*).

We did not collect physiological measures of stress or information about the COVID-19 infection status of participants, which precludes a direct exploration of the immediate effects of experiencing the infection on belief-updating behavior and the potential interaction with anxiety and stress levels. Although subjective ratings of the perceived risk of death from COVID-19 correlated negatively to

the beliefs updating bias measured during the pandemic, this result was obtained in a subset of participants, retrospectively (Appendix 5). We thus cannot directly attribute the observed lack of optimistically biased belief updating during the lockdown to psychological causes such as heightened anxiety and stress. This limitation is noteworthy, as the impact of experiencing the pandemic on belief updating about the future could differ between those who directly experienced infection and those who remained uninfected. It is also important to acknowledge that our study was timely and geographically limited to the context of the COVID-19 outbreak in France. Cultural variations and differences in governmental responses to contain the spread of SARS-CoV-2 may have impacted the optimism biases in belief updating differently.

The observed lack of optimistically biased belief updating may be interpreted as an adaptive response to the experience of an unprecedented level of uncertainty and chronic threat during a global crisis. Although we did not have access to anxiety and stress perceptions during and outside the pandemic, our computational modeling results corroborate to some extent this interpretation. Notably, belief updating was more optimistically biased RL-like outside the pandemic and more rational Bayesian-like during the pandemic. The biased RL-like updating behavior observed outside the pandemic indicated that participants relied on the motivational salience of positive estimation errors to teach them how to update their beliefs about the future by trial and error. This finding aligns with past work, showing that RL-like updating models best explain belief updating in non-threatening, non-stressful, and predictable laboratory contexts (*Kuzmanovic and Rigoux, 2017*). It further suggests that RL strategies are a computationally efficient way to guide decision-making and belief formation when the environment is stable and predictable (*Gershman and Daw, 2017*). For instance, in environments with well-defined reward structures, the human brain has been shown to efficiently rely on RL and avoid a computational overhead associated with the Bayesian-like inference process (*Daw et al., 2005*). On the contrary, belief updating was more rational Bayesian-like during the COVID-19 pandemic, indicating that participants weighed the uncertainty of evidence in favor of and against their prior beliefs. This finding aligns with research about Bayesian networks to model semantic knowledge processing under uncertainty (*Pearl, 1988*) and with work that uses Bayes rule to understand how humans learn and choose under uncertainty (*Tenenbaum et al., 2011*; *Griffiths and Tenenbaum, 2006*; *Gershman et al., 2015*).

It is essential to acknowledge that computational modeling provides insight into potential mechanisms, but this excludes inferences on whether humans indeed update beliefs in the way the best-fitting model assumes. Other models, such as evidence accumulation models, may also work when humans update their beliefs about the future and their immediate surroundings (*Gesiarz et al., 2019*). Unfortunately, we did not assess reaction times during belief updating, which is crucial for fitting evidence accumulation models such as drift-diffusion models to observed behavior. However, we can infer from our findings that the two model families employed to fit observed belief-updating behavior represented two different but complementary prediction strategies. These strategies were then used to function in the uncertainty of real-life conditions. We call for more studies investigating these computational models' psychological and biological validity under certainty and uncertainty.

In this study, we tested how actual adverse experiences affect the updating of negative future outlooks in healthy participants and in analogy to studies conducted in depressed patients (*Garrett et al., 2014*; *Korn et al., 2014*; *Bottemanne et al., 2022*) following the cognitive model of depression (*Beck et al., 1979*). One open question is whether findings were specific to the adverse event framing (*Harris and Hahn, 2011*; *Shah et al., 2016*; *Burton et al., 2022*). We argue that under normal, non-adverse contexts, belief updating should also be optimistically biased for positive life events, as shown by previous research (*Marks and Baines, 2017*; *Garrett and Sharot, 2017*). However, how context such as experiencing a challenging or favorable situation influences the updating of beliefs about positive and negative outlooks remains an open question.

In conclusion, our results provide insight into the resilience and adaptability of belief-updating processes during and following the COVID-19 pandemic. They demonstrate the malleability of the human ability to anticipate the future and how it can adapt to real-life conditions under which an overly optimistic view of future risks would be harmful.

**Table 1.** Sociodemographic data for all four groups (N=123).♀: Female; ♂: Male; Note: education is the number of years completed in higher education after a high school diploma.

|  | October 2019 (N=30) | March – April 2020 (N=34) | May 2021 (N=31) | June 2022 (N=28) |
|---|---|---|---|---|
| Age (years) | 34±2 | 42±3 | 42±3 | 35±3 |
| Gender | 18 ♀, 12 ♂ | 25 ♀, 9 ♂ | 20 ♀, 11 ♂ | 14 ♀, 14 ♂ |
| Education (years) | 5±0.4 | 4±0.3 | 5±0.2 | 4±0.4 |

## Methods

### Ethical considerations

The Local Ethics Committee of Sorbonne University approved the study. All participants provided informed consent and consent to publish. The study protocol followed the Declaration of Helsinki. The authors declare no competing interests.

### Participants

One hundred twenty-five participants (mean age = 37.50 ± 1.28, 99 females) allotted to four different groups were recruited for the study (see *Table 1*; *Appendix 7—table 15*) via a public advertisement. Two participants from the group tested in June 2022 were excluded from the analyses because they always indicated the same risk estimate for each event.

### Experimental design

The first group of 30 participants (mean age = 33.73 ± 1.96, 18 females) was recruited in October 2019 before the COVID-19 outbreak in France (*Figure 4a*). These participants were tested in the laboratory. A second group of 34 participants (mean age = 42.24 ± 3.34, 25 females) was recruited from March to April 2020 for online testing during the first COVID-19-related lockdown of social and economic life, with schools closed. A third group of 31 participants (mean age = 42.42 ± 3.35, 20 females) was recruited and tested online immediately after the last lockdown and still during the COVID-19 pandemic in May 2021. A fourth group of 30 participants (mean age = 34.66 ± 2.71, 16 females) was recruited at the lift of the COVID-19 pandemic-related state of emergency and tested in the laboratory in June 2022 (*Figure 4a*). This group was also used to rule out an eventual effect of task design. Half of them (n=15) performed a one-run task design, and the other half (n=15) performed a two-run task design (e.g. see in more detail the belief updating task description below). Note the 30 participants tested before the COVID-19 pandemic were recontacted during the first strict lockdown to re-perform the belief updating task online (*Figure 4a*). This allowed us to check for the effects of experiencing a COVID-19-related lockdown within the same cohort of participants. Two of the 30 participants in this group did not respond. Therefore, the sample size for the within-group test-retest analyses was 28 participants.

### Sample sizes

The sample sizes were determined by a power analysis using the *power curve* function in R (version 1.2.5033) and building on the good news/bad news bias observed in the first group tested in October 2019 before the COVID-19 outbreak in France. The sample size required to replicate a significant effect of estimation error valence on the updating with a power between 80% and 90% lay between 28 and 35 participants, respectively.

### Belief updating task

All participants performed a belief-updating task (*Figure 4b*). For in-person testing, stimulus presentation and response recording were done with the Psychophysics toolbox in MATLAB (R2018b, Update 6, version 9.5.0.1265761). The online testing was done using Qualtrics (Qualtrics Software, version March 2020 of Qualtrics, Copyright 2020 Qualtrics. Available at https://www.qualtrics.com).

The task comprised 40 trials with 40 adverse lifetime events and base rates. In each trial, participants were asked to estimate the likelihood of experiencing an adverse event in the future for themselves

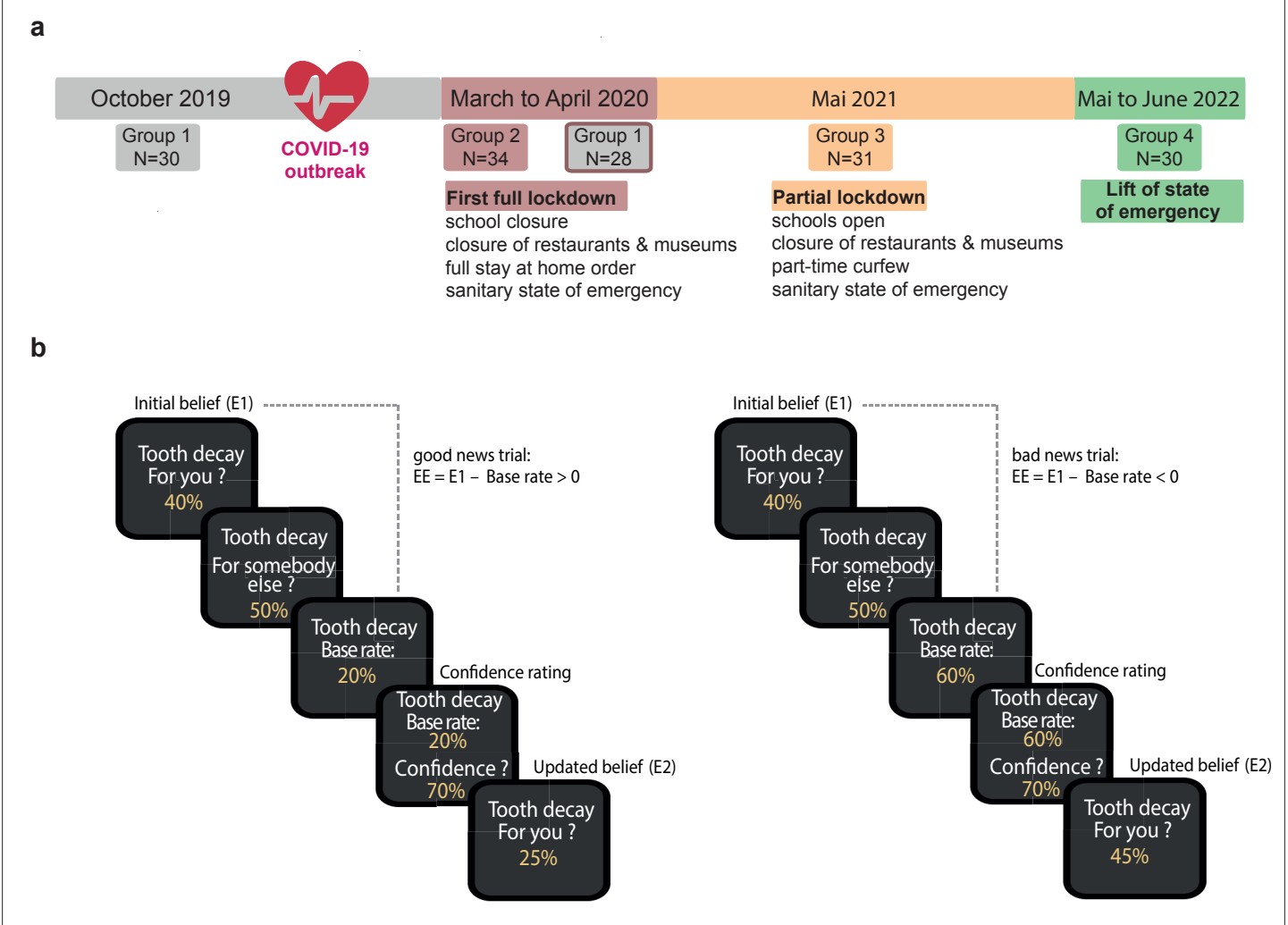

**Figure 4.** Experimental design. (**a**) Timeline of testing. Four groups were tested, before the COVID-19 outbreak in October 2019, during the first complete lockdown of social and economic life in March and April 2020, after a partial lockdown in May 2021, and after the lift of the pandemic-related state of emergency in June 2022. (**b**) *Belief updating task.* Panels show subsequent appearances on the screen within a good news trial (left panels) and a bad news trial (right panel). Responses were self-paced. The task goal was to estimate the risk of experiencing different adverse future life events (e.g. tooth decay) for oneself (E1) and for somebody else (eBR) before and after (E2) being presented with information about the event's prevalence in the general population (i.e. base rate (BR)).

and somebody else before and after receiving information about the likelihood of occurrence in the general population (i.e. the base rate). The adverse life events and their actual base rates were taken from previously published work in healthy controls (*Sharot et al., 2011*; *Sharot and Garrett, 2022*). The base rates for events were normal to uniformly distributed (W=0.952, p=0.088, Shapiro-Wilk test). The base rates ranged between 10% and 70%, with a mean of 40%. Participants rated their estimates between 3% and 77%, which ensured that for most likely (base rate = 70%) and most unlikely events (base rate = 10%) there was enough space (7%) to update beliefs toward the base rates (*Garrett and Sharot, 2017*; *Sharot and Garrett, 2022*). Moreover, all statistical models included the absolute estimation errors as a control for variance potentially explained by different estimation error magnitudes (*Garrett and Sharot, 2017*; *Sharot and Garrett, 2022*).

In more detail, as illustrated in *Figure 4b*, each of the 40 trials began with presenting an adverse life event. Participants estimated their own risk and subsequently the risk of someone else their age and gender. Then the base rate of the event occurring in the general population was displayed on the computer screen. Participants rated their confidence in the accuracy of the presented base rate. Finally, they re-estimated their risk of experiencing the event now informed by the base rate.

The task design varied between some groups:

Fifty-eight participants underwent assessment outside the COVID-19 pandemic, with 45 performing a two-run task design (n=30 tested before the outbreak in October 2019; n=15 tested at the end of the sanitary state of emergency in June 2022). The remaining 13 participants tested outside the pandemic performed the one-run task design like the 65 participants tested during the pandemic.

In the two-run task design, participants performed a first run of 40 trials. Each trial started with the display of an adverse lifetime event. Participants were asked to estimate the risk of experiencing this event in the future for themselves (E1 rating) and for somebody else (eBR rating). At the end of each trial, they received information about the event's base rates and rated their confidence. In a second run, they saw an adverse future life event and its base rate on each trial. They then re-estimated their risk (E2 rating) on a trial-by-trial basis.

The one-run task design is displayed in *Figure 4b* and consists of one run of 40 trials. Within each trial, participants first estimated the risk of experiencing a future adverse lifetime event for themselves (E1 rating) and for somebody else (eBR rating), were presented with the base rate for this event (BR), rated their confidence in the base rate and re-estimated their risk of experiencing the event in the future (E2 rating). Note that all analyses were controlled for these differences in task design, which had non-significant effects on belief updating, confidence ratings, estimation error magnitude, and learning rates (see corresponding tables in Appendix 7 with LME results).

## Belief updating task measures of interest

The estimation error indicated whether participants overestimated or underestimated their likelihood of experiencing an adverse event (E1) relative to its actual base rate (BR). The estimation error (EE) was calculated according to the equation i:

$$EE = E1 - BR \tag{i}$$

The estimation error was further used to categorize trials into good or bad news trials:

For good news trials, the estimation error was positive (EE >0), which indicated an overestimation of one's likelihood of experiencing an adverse life event relative to the base rate of that event (E1 >BR). For bad news trials, the estimation error was negative (EE <0), which indicated an underestimation of one's likelihood of experiencing an adverse event relative to its actual base rate (E1 <BR).

The main variable of interest was the magnitude of belief updating (UPD), which was calculated as the difference between the first (E1) and the second (E2) estimate after receiving information about the base rate (BR). Notably, the update was calculated for good and bad news trials, respectively, following equation ii:

$$
\begin{aligned}
&if\,EE > 0\\
&\qquad UDP_{good\,news} = E1 - E2\\
&if\,EE < 0\\
&\qquad UDP_{bad\,news} = E2 - E1
\end{aligned}
\tag{ii}
$$

Lastly, the difference between updating after good and bad news was calculated to assess the updating bias following equation iii:

$$UDB = UPD_{good\,news} - UPD_{bad\,news} \tag{iii}$$

A positive difference indicated that participants updated their beliefs about their lifetime risk of experiencing adverse life events more frequently following good news than bad news.

For each participant, trials that did not receive a response (on average 0.44 trials per subject) and trials with an EE = 0 (on average 0.63 trials per subject) were excluded from the analyses.

The distance measured the extent to which participants consider their probability of experiencing a given adverse event (E1) different from the lifetime risk of someone from a similar socio-economic background (eBR). If positive, it reflected an optimistic bias in initial estimates. The following, distance = eBR – E1, was calculated. Additional analysis to control for this measure was added (*Appendix 7— table 3*).

## Model-free statistical analyses of observed belief updating behavior

The main aim of this study was to assess how belief updating was affected by the context of experiencing the COVID-19 pandemic.

We, therefore, conducted between-context analyses, contrasting groups tested during (i.e. during the first lockdown in March/April 2020 and immediately after the last lockdown in May 2021) and outside the pandemic context (i.e. before the outbreak in October 2019 and 1 year after the last lockdown in June 2022). All statistical tests were conducted using the MATLAB Statistical Toolbox (MATLAB 2018b, MathWorks) and JASP (JASP 0.16.4).

A first linear mixed effects model (LME 1) was fitted to the belief updating, following equation *iv*:

$$\text{UPD} = \beta_0 \text{ Intercept } + \beta_{\text{EE}} |\text{EE}| + \beta_{\text{EE}}\text{valence} + \beta_{\text{context}} \text{ context} + \beta_{\text{design}}$$
$$\text{design} + \beta_{\text{age}} \text{ age} + \beta_{\text{gender}} \text{ gender} + \beta_{\text{education}}\text{education} + \beta_{\text{EE valence} * \text{context}}$$
$$\text{EE valence by context} + (1 \mid \text{subject}) + (1 + |\text{EE}| \mid\text{subject}) + (1 + \text{EE valence} \mid$$
$$\text{subject})$$

(iv)

The model included fixed effects for estimation error magnitude ($|\text{EE}|$), estimation error valence (EE valence, coded –1 for bad news trials and 1 for good news trials), context (coded 0 for outside and 1 for during the COVID-19 pandemic), task design (coded 1 for one-run, 2 for two-run design), age, gender (coded 0 for male, 1 for female participants), level of education, and the interaction of interest EE valence by context. The model also included random intercepts nested by subject number and random slopes for estimation error magnitude and valence.

Subsequently, the same Linear Mixed Effects (LME) model was applied again to the belief update to explore the categorical effect of context in conjunction with EE valence. This allowed for a more specific comparison of the impact of EE valence between contexts (groups): those tested before the COVID-19 pandemic outbreak in October 2019 (baseline) compared to those tested during the initial COVID-19-related lockdown (context 1), those tested immediately after the last lockdown during the pandemic in May 2021 (context 2), and those tested one year post-pandemic in June 2022 (context 3), respectively.

Post-hoc two-tailed and one-tailed t-tests were conducted to characterize the directionality of detected main effects and interactions.

## Posthoc power analysis

The best fitting computational models of belief updating in each context (i.e. during and outside the pandemic) and their free parameters were used to simulate new belief updates (*Wilson and Collins, 2019*). Simulations were repeated 1000 times. At each iteration, the above-described linear mixed effects model (equation iv) was fitted to the simulated belief updates. The frequency across 1000 iterations with which the LME detected a significant interaction of valence by context on simulated belief updating indicates the power of this interaction effect and the chance for type II errors of failing to reject the null hypothesis when the effect was there.

## Model-based analyses of belief-updating behavior

To gain more insight into putative cognitive mechanisms of belief updating during and outside the COVID-19 pandemic, two families of non-linear computational models were fitted to observed belief updating behavior, which is specified below.

## Model specifications
### Reinforcement learning model of belief updating

A Reinforcement learning-like model assumed that belief updating is proportional to the magnitude of the estimation error following *Kuzmanovic and Rigoux, 2017*. The learning rate scaled the effect of the estimation error on belief updating following the generic equation v:

$$\text{UPD} = \text{LR} * \text{EE}$$

(v)

Importantly, the learning rate was estimated for good and bad news trials separately and following equations *vi* and *vii*:

$$\text{LR}_{\text{good news}} = S + A \tag{vi}$$

$$\text{LR}_{\text{bad news}} = S - A \tag{vii}$$

For both types of trials, the learning rate was composed of two components that varied across participants. The scaling parameter (S) measured the extent to which a participant took the estimation error into account when updating beliefs. The asymmetry parameter (A) indicated to what extent the belief updating differed for positive and negative estimation errors. The priors for scaling and asymmetry were untransformed and unbound. The mean of the prior distribution for scaling was set to one. Thus, a scaling of one meant that the updating magnitude equaled the estimation error magnitude. The mean of the prior distribution for the asymmetry parameter was set to zero. An asymmetry parameter value larger than zero meant positively biased updating, whereas an asymmetry parameter smaller than zero meant negatively biased belief updating.

A version of the RL-like model of belief updating took the personal relevance (PR) of presented adverse future life events into account following equations viii and ix:

$$\text{UPD}_{\text{good news}} = (S + A) * \text{EE} * (1 - \text{PR}) \tag{viii}$$

$$\text{UPD}_{\text{bad news}} = (S - A) * \text{EE} * (1 - \text{PR}) \tag{ix}$$

The PR weighed the estimation error (EE) and corresponded to the difference between the estimated base rate (eBR, the estimated risk for somebody else) and the initial estimate (E1, the estimated risk for oneself). Based on the sign of this difference between eBR and E1, the PR was calculated following equations x to xii:

$$\textit{If } eBr < E1$$
$$PR = \frac{eBR - E1}{eBR - 1} \tag{x}$$

$$\textit{If } eBR > E1$$
$$PR = \frac{E1 - eBR}{99 - eBR} \tag{xi}$$

$$\textit{If } eBR = E1$$
$$PR = 0 \tag{xii}$$

## Bayesian belief updating model

A second family of computational models was fitted to belief updating behavior and assumed that belief updating was proportional or equal to the Bayes rule, following equations xiii and xiv (***Kuzmanovic and Rigoux, 2017***):

$$\text{UPD}_{\text{good news}} = (E1 - E2b) * (S + A) \tag{xiii}$$

$$\text{UPD}_{\text{bad news}} = (E2b - E1) * (S - A) \tag{xiv}$$

The scaling parameter (S) corresponded to the tendency of participants to update their beliefs in response to the presented base rate following Bayes' rule. A scaling smaller than one (S<1) indicated lesser belief updating than what was predicted by the Bayes rule, and a scaling larger than one (S>1) indicated more updating than predicted by the Bayes rule.

The Bayes rule was used to define a Bayesian second estimate (E2b, the updated belief), which was calculated following equations xv and xvi:

$$E2b = \text{Prior} * \text{LHR} \tag{xv}$$

$$E2b = \frac{\text{Prior} * \text{LHR}}{1 + (\text{Prior} * \text{LHR})} \tag{xvi}$$

With the Prior=P(BR), corresponding to the base rate (BR) of each event following equation xvii:

$$Prior = \frac{\text{BR}}{1 - \text{BR}} \tag{xvii}$$

The Likelihood Ratio (LHR) indicates the probability of the initial estimate (E1) relative to the likelihood of the alternative estimated base rate (eBR) following equation xviii:

$$LHR = \frac{\dfrac{E1}{1 - E1}}{\dfrac{eBR}{1 - eBR}} \qquad \text{(xviii)}$$

Alternative models of these two model families (RL and Bayesian) were fitted to the observed belief-updating behavior. Each model alternative represented a different combination of free parameters composing the learning rate to test a total of 12 assumptions about the cognitive process underlying belief updating:

RL model 1. Belief updating is asymmetrical and proportional to the estimation error: S+A varied across participants.

RL model 2. Belief updating is non-asymmetrical and proportional to the estimation error: S varied, A was silent (fixed to zero).

RL model 3. Asymmetrical updating is equal to the estimation error: S was fixed (to one), and A varied.

RL model 4. Updating equals the estimation error: S and A were fixed.

RL model 5. Belief updating is asymmetrical, proportional to the estimation error, and moderated by the personal relevance of events (PR): S+A varied. PR was weighting the EE following equations x., xi., and xii.

RL model 6. Belief updating is non-asymmetrical and proportional to the estimation error moderated by PR: S varied, A was fixed, and PR weighted the EE following equations x., xi., and xii.

RL model 7. Asymmetrical updating equals the estimation error moderated by PR: S was fixed, A varied, and PR weighted the EE following equations x., xi., and xii.

RL model 8. Updating equals the estimation error moderated by PR: S and A were fixed, and PR weighted the EE following equations x., xi., and xii.

Bayesian model 1. Belief updating is asymmetrical and proportional to Bayes rule: S and A varied.

Bayesian model 2. Belief updating is proportional to a rational Bayes rule: S varied, and A was fixed.

Bayesian model 3. Belief updating equals an asymmetrical Bayes rule: S was fixed, and A varied.

Bayesian model 4. Belief updating equals a rational Bayes rule: S and A were fixed.

## Model estimation

Models were estimated following the procedure reported by *Kuzmanovic and Rigoux, 2017* and *Bottemanne et al., 2022*. In short, models were not hierarchical, and parameter estimation was thus less sensitive to differences in group sample sizes. For each participant, optimal scaling and asymmetry parameter values were obtained using Bayesian variational inferences implemented in the VBA toolbox (*Daunizeau et al., 2014*).

## Model comparisons

The free energy approximations for a model's evidence in each participant were entered into a random effect Bayesian model comparison that yielded the two criteria considered for model selection: the estimated model frequency (Ef) in each group (estimate the frequency of each model in the population) and the protected exceedance probability (pxp), which corresponded to the probability that the hypothesis predominates in the population, above and beyond chance.

## Parameter recovery

Parameter recovery analysis was conducted to check whether the free parameters of the winning models were identifiable and described the data better than any other set of parameters. The procedure was the same as reported in *Bottemanne et al., 2022*. In short, to validate the accuracy of the fitting procedure in providing meaningful parameter values, simulated belief updating data was generated using the observed parameter values for both the optimistic RL model and the optimistic Bayesian updating model. Subsequently, we applied the fitting procedure to these simulated data to iteratively 'recover' the parameters. Thereby, the means of the parameters were set to correspond to

the observed sample means (i.e. scaling = 0.39 ± 0.02, asymmetry = 0.07 ± 0.01 for the RL model; scaling = 0.42 ± 0.03, asymmetry = 0.05 ± 0.01 for the Bayesian model). This process was iterated to simulate 40 values of belief updates 123 times. The model was then inverted by fitting it to the simulated data, yielding a new set of recovered values for scaling and asymmetry. Finally, the recovered and estimated parameters were compared by assessing their correlation using Pearson's correlation coefficients.

## Parameter comparisons

To compare learning rates and learning rate components across groups, we used the parameters from the optimistically biased RL-like model (RL model 1), which performed best when fitted to the whole dataset (Ef = 0.40, pxp = 0.99) and reproduced the observed updating behavior as shown in *Figure 2—figure supplement 3*.

Individual learning rates from this RL model 1 and their scaling and asymmetry components were the dependent variables (DV) of the following generic linear mixed effects model (equation xix and xx):

$$LR = \beta_0\text{Intercept} + \beta_{\text{valence}}\text{ valence} + \beta_{\text{context}}\text{ context} + \beta_{\text{design}}\text{ design} + \beta_{\text{age}}\text{ age} + \beta_{\text{gender}}$$
$$\text{gender} + \beta_{\text{education}}\text{ education} + \beta_{\text{valence}}\text{ valence by context} + (1|\text{subject}) \tag{xix}$$

$$parameters = \beta_0\text{Intercept} + \beta_{\text{context}}\text{ context} + \beta_{\text{age}}\text{ age } \beta_{\text{gender}}\text{ gender } \beta_{\text{education}}$$
$$\text{education} + (1|\text{subject}) \tag{xx}$$

The model included fixed effects for news valence (valence, coded 1 for good news, –1 for bad news), context (coded 0 for outside the pandemic, 1 for during the pandemic), task design (coded 1 for one-run and 2 for two-run), age, gender (coded 0 for male, 1 for female participants), and level of education. It also tested the interaction of interest context by valence. The intercept was nested at random by subject number.

Post-hoc one-sampled and two-sampled t-tests were conducted to characterize the directionality of effects.

## Model recovery

To check if the 12 models were identifiable, a model recovery analysis was conducted using the VBA toolbox (*Daunizeau et al., 2014*). In more detail, behavior was simulated using each of the 12 models with parameters estimated from participants' actual behavior. These simulated datasets were then refitted to all alternative models. The model comparison procedure was then performed to evaluate whether each model could accurately recover the parameters that generated the data. This resulted in a 12x12 confusion matrix that compared the performance of all models in fitting each simulated dataset (*Figure 2—figure supplement 2*). The matrix shows the estimated frequency when fitting to the 12 models (y axis), the behavior generated by each model (x axis), and provided evidence for strong recovery of nearly all models and, importantly, the two winning models: the optimistically biased RL-like model and the rational Bayesian model of belief updating. This analysis thus rules out that the two model families were confused and mitigates concerns about the validity of the model selection.

# Acknowledgements

We thank Tali Sharot for helpful comments on the results. This work was supported by core funding from the Paris Brain Institute Foundation.

# Additional information

## Funding

| Funder | Grant reference number | Author |
| --- | --- | --- |
| Agence Nationale de la Recherche | ANR-21-CE37-0014 | Liane Schmidt |

The funders had no role in study design, data collection and interpretation, or the decision to submit the work for publication.

## Author contributions

Iraj Khalid, Data curation, Software, Formal analysis, Investigation, Visualization, Methodology, Writing – original draft, Writing – review and editing; Orphee Morlaas, Conceptualization, Data curation, Formal analysis, Investigation, Methodology; Hugo Bottemanne, Conceptualization, Data curation, Investigation, Questionnaire development; Lisa Thonon, Data curation, Software, Formal analysis, Investigation; Thomas Da Costa, Data curation, Formal analysis, Investigation; Philippe Fossati, Conceptualization, Supervision, Validation, Writing – review and editing; Liane Schmidt, Conceptualization, Resources, Data curation, Software, Formal analysis, Supervision, Funding acquisition, Validation, Investigation, Visualization, Methodology, Writing – original draft, Project administration, Writing – review and editing

## Author ORCIDs

Iraj Khalid ⓘ https://orcid.org/0000-0002-5396-5055
Hugo Bottemanne ⓘ https://orcid.org/0000-0003-2958-0849
Thomas Da Costa ⓘ https://orcid.org/0009-0008-6983-6259
Liane Schmidt ⓘ https://orcid.org/0000-0002-4159-9705

## Ethics

The Local Ethics Committee of Sorbonne University approved the study. All participants provided informed consent and consent to publish. The study protocol followed the Declaration of Helsinki. The authors declare no competing interests.

Reviewer #1 (Public review): https://doi.org/10.7554/eLife.101157.3.sa1
Reviewer #2 (Public review): https://doi.org/10.7554/eLife.101157.3.sa2
Author response https://doi.org/10.7554/eLife.101157.3.sa3

---

# Additional files

## Supplementary files

MDAR checklist

## Data availability

Source data files to generate the figures and figure supplements have been provided.

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

## Appendix 1

### Within-group comparisons

Belief updating was compared within a group of participants (n=28), who were tested both before (October 2019) and during the COVID-19 pandemic (March-April 2020). A mixed effects linear regression analysis of belief updating showed a significant main effect of EE valence ($\beta$=4.05, SE = 1.22, t(103) = 3.31, p=0.001, 95% CI [1.63–6.47]), as well as a significant effect of testing context ($\beta$=–4.41, SE = 1.49, t(103) = –2.95, p=0.004, 95% CI [-7.37—1.45]), and more importantly a significant EE valence by testing context interaction ($\beta$=–7.66, SE = 1.49, t(103) = –5.13, p<0.001, 95% CI [-10.62 to –4.70], *Figure 1—figure supplement 1*, *Appendix 7—table 4*). A significant main effect of gender was also found ($\beta$=3.53, SE = 1.64, t(103) = 2.15, p=0.03, 95% CI [0.27–6.79]), with women updating their beliefs more than men. Post-hoc t-tests revealed that participants tested before the emergence of the pandemic updated their beliefs more after receiving good news (mean $UPD_{good}$ = 14.38 ± 1.14) than after receiving bad news (mean $UPD_{bad}$ = 6.23 ± 1.17; t(27) = 4.93, p<0.001, Cohen's d=0.93, paired sample, two-tailed t-test). This asymmetry in belief updating was not observed when the same 28 participants were tested during the first lockdown (mean $UPD_{good}$ = 2.25 ± 1.76; mean $UPD_{bad}$ = 9.30 ± 2.47; t(27) = –1.84, p=0.08, Cohen's d=–0.35, paired sample, two-tailed t-test).

Bayesian model comparison in the group of participants tested both before and during the lockdown showed that belief updating was more RL-like and optimistic before the lockdown (Ef = 0.73, pxp = 1), and rational Bayesian-like during the lockdown (Ef = 0.61, pxp = 0.99; *Figure 2—figure supplement 1*).

## Appendix 2

### Optimism bias in initial beliefs

As shown in *Figure 1—figure supplement 2*, the participants displayed an optimism bias in their initial belief estimates: They estimated that adverse events are more likely to happen to others than to themselves ($\beta$=3.02, SE = 0.86, t (232)=3.53, p=0.001, 95% CI [1.33–4.71]; *Appendix 7—table 18*). However, the pandemic context had no significant effect ($\beta$=–1.91, SE = 3.00, t (232)=–0.64, p=0.52, 95% CI [-7.82–4.00]; *Appendix 7—table 18*). We conclude from these additional analyses that the pandemic context specifically influenced optimistically biased belief updating but did not affect optimism bias in initial beliefs.

Note that optimism bias is the propensity to believe that negative events are more likely to happen to others than to oneself. In contrast, the terms optimistically biased belief updating or optimism biases in belief updating used throughout the main text refer specifically to the valence effect in belief updating —updating beliefs more in response to good news than to bad news.

# Appendix 3

## Sources of variance in belief updating

In *Figure 1b*, participants tested during the COVID-19 pandemic showed more variance in belief updating in response to both good and bad news than those tested outside the pandemic. This variability might be because they ignored the base rates when updating their beliefs, possibly influenced by the shift to online testing during the pandemic.

If participants paid attention to the base rates, they were expected to update toward the base rate, yielding positive values for the update. On the contrary, ignoring the base rates can be reflected by updating away from the base rate, with second estimates (E2) that, in the case of good news trials, lay above the first estimate (E1; e.g., E1=60%, BR = 40 %, E2=70 %, UPD = E1 - E2 = - 10 %). Likewise, in the case of bad news trials, second estimates may lie below the first estimate (e.g. E1=20 %, BR = 40 %, E2=10 %, UPD = E2 - E1 = –10%). We examined the number of trials with such paradoxical second estimates yielding negative values for the belief update. We found no significant difference (t(121) = 1.77, p=0.08, Cohen's d=0.32, two-sample two-tailed t-test) between the number of paradoxical trials in participants tested outside (6.09±0.47 trials on average) and during (4.77±0.56 trials on average) the pandemic. This suggests that participants tested online exhibited no greater propensity for paradoxical responses than those tested in person (non-significant effect of context on the number of paradoxical trials: $\beta$=–0.19, SE = 1.57, t(234) = –0.12, p=0.90, 95% CI [-3.28–2.90]; *Appendix 7—table 16*).

Second, we tested if the observed variance in belief updating between the groups was due to second estimates that over- and undershot the base rates. For instance, individuals who undershot indicated second estimates below base rates signaling good news (e.g. E1=40 %, BR = 20 %, E2=10 %). In contrast, individuals who overshot indicated second estimates above the base rates signaling bad news (e.g. E1=10%, BR = 20%, E2=40%). Undershooting might indicate an attuned sensitivity to good news and overshooting an attuned sensitivity to bad news. We found that participants tested outside the COVID-19 pandemic undershot more often (on average 4.48±0.46 trials) than they overshot (on average 2.38±0.33 trials; t(57) = 3.76, p<0.001, Cohen's d=0.49, paired-sample, two-tailed t-test). This propensity aligned with the bias to update beliefs more after good than after bad news in this group. Conversely, participants tested during the pandemic didn't show a significant difference in the number of trials where they under- (2.00±0.32 trials on average) or overshot (3.08±0.73 trials on average; t(64) = –1.34, p=0. 19, Cohen's d=–0.17, paired-sample, two-tailed t-test), aligning with the absence of the good news/bad news bias in belief updating. Critically, when comparing the two groups, we found a significant positive interaction testing context by type of shooting ($\beta$=1.66, SE = 0.50, t(233) = 3.33, p=0.001, 95% CI [0.68–2.65]; *Appendix 7—table 17*). Post-hoc t-tests showed that participants tested outside the COVID-19 pandemic undershot more often than participants tested during the pandemic (t(121) = 4.48, p<0.001, Cohen's d=0.81, two-sample two-tailed t-test). No differences were observed between the groups regarding overshooting (t(121) = –0.84, p=0.40, Cohen's d=–0.15, two-sample two-tailed t-test). These findings indicated that participants tested outside the pandemic were more sensitive to good news, corroborating the findings of a more robust good news/bad news bias in belief updating found in this group.

# Appendix 4

## Parameter and model recovery

### Parameter recovery for the winning model in each group of participants

To ensure that parameter estimates were robust for the best-fitting models within each context (i.e. during and outside the pandemic), we conducted parameter recovery in each group of participants. For the participants tested outside the pandemic scaling and asymmetry parameters were strongly recovered by the RL-like model (Model 1; *Figure 3—figure supplement 1*, scaling: $r=0.92$, p=0.001; asymmetry: $r=0.75$, p<0.001). For the group tested during the pandemic recovery for the scaling ($r=0.85$, p<0.001) and asymmetry ($r=0.71$, p<0.001) parameters by the Bayesian model of belief updating was good (*Figure 3—figure supplement 1*).

### Model recovery and confusion analysis of the computational models of Belief Updating

To ensure the robustness and specificity of our computational models, we performed a model recovery analysis. Using simulated data, we tested whether each of the 12 models in our comparison framework could be correctly identified under the estimation and selection criteria used in our study. Results indicated that all models were well-recovered (*Figure 2—figure supplement 2*), except for models in both families that use fixed parameters (i.e. scaling = 1 and asymmetry = 0), which were better recovered by the model with the scaling parameter fixed but the asymmetry parameter estimated iteratively.

### Comparison of observed and modeled behavior

To check if the overall best fitting optimistically biased RL model reproduced observed belief updating behavior, we compared the observed belief updates in each participant to the updates predicted by the overall winning model (model 1 – RL model with both parameters estimated). This comparison is depicted in *Figure 2—figure supplement 3*, which highlights a strong correspondence between the actual data and the modeled estimates.

# Appendix 5

## Correlations between self-reported attitudes during the COVID-19 lockdown and belief updating

Some self-reported anxiety and perceived risk measures experienced during the lockdown were collected in a subset of participants (n=40, counting n=21 tested both before and during the 1st strict lockdown, and n=19 tested solely during the 1st lockdown). These reports were given retrospectively at the time of the release of the 1st lockdown in summer 2020 when the pandemic was still unfolding (*Appendix 7—table 1*).

Exploratory correlations revealed some noteworthy trends, which though did not survive corrections for multiple comparisons. We found that participants who reported having perceived a bigger risk of death due to contagion were also those who were less optimistically biased when updating their beliefs about adverse future life risks during the first strict COVID-19-related lockdown ($r$=–0.36, p=0.02, Pearson's correlations).

Moreover, parameter estimates from the computational models of belief updating showed associations with specific survey responses: The Bayesian model's scaling parameter correlated positively with adherence to distancing measures ($r$=0.41, p=0.01) and negatively with the need for social contact ($r$=–0.37, p=0.02). This result indicated that participants who were updating their beliefs faster were more likely to follow preventive guidelines and felt less social craving. Meanwhile, the asymmetry parameter correlated negatively with mask wearing ($r$=–0.41, p=0.01), positively with physical contact with close others ($r$=0.32, p=0.04) and satisfaction with social interactions ($r$=0.33, p=0.04). This suggests that participants who displayed some asymmetry in belief updating during the COVID-19 pandemic were less likely to comply with mask-wearing rules and more likely to engage in social interactions.

Note that the correlations between these questionnaire measures of behavior during COVID-19 related lockdowns and the free parameters hold for both model families (RL and Bayesian). However, for the sake of parsimony, we reported only correlations to the parameters from the Bayesian model, as it provided the best fit to our observed belief updating behavior during the COVID-19 period.

## Appendix 6

### Belief updating task instructions

The following instructions (translated from French) were displayed to participants prior to performing the task:

This task measures your beliefs about future life events. You will be presented with 40 different future life events, one at a time. At each time you are asked to estimate:

1. The likelihood of the event happening in your future life.
2. The likelihood of the event happening to someone else your age and gender.
   You will then see the base rate for the event and will be asked to:
3. Rate your confidence in the base rate information – how much you believe it to be accurate on a scale between 0 and 100%.
   At the end of each trial and after having seen the base rate for the event, you are again asked to estimate:
4. The likelihood of the event happening in your future life.

Please estimate the likelihoods of the future life events on a scale between a minimum likelihood of 3% and a maximum likelihood of 77%.

## Appendix 7

**Appendix 7—table 1.** Survey responses in n=40 participants tested during the pandemic.

| Category | Specific question | Mean | sem |
|---|---|---|---|
| | COVID-19 risk | 2.9 | 0.2 |
| | General risk perception | 1.8 | 0.1 |
| Risk perception | COVID-19 mortality risk | 3.1 | 0.2 |
| | Mask wearing | 3.7 | 0.2 |
| | Social distancing outside home (shops, work) | 4.4 | 0.1 |
| | Hand washing | 4.2 | 0.2 |
| | Social distancing at home | 4.0 | 0.1 |
| | Gloves wearing | 1.4 | 0.1 |
| | Shaking hands, hugging | 2.2 | 0.2 |
| | Leave home for work, errands | 3.7 | 0.1 |
| Adoption of protective measures | Hand sanitizer use | 4.0 | 0.2 |
| | Social craving | 3.8 | 0.2 |
| | Feeling isolated | 2.6 | 0.2 |
| | Calling friends, parents, family, acquaintances | 4.5 | 0.1 |
| | Losing contact with friends, acquaintances | 2.3 | 0.2 |
| | Social media use | 3.6 | 0.2 |
| | Feeling of isolation from loved ones | 2.9 | 0.2 |
| Need for social interaction | Quality of social interactions | 3.9 | 0.1 |
| Mood | Sadness and anxiety | 2.5 | 0.2 |
| Anxiety | Level of anxiety | 2.6 | 0.2 |
| Living | 1 – alone, 2 – alone with pet, 3 – couple w/o children, 4 – couple w/ children, 5 – Family | 3.3 | 0.2 |
| Housing | 1 – apartment, 2 – apartment w/ outdoor space, 3 – house, 4 – house w/ outdoor space | 2.3 | 0.2 |
| Occupation | 1 – no occupation. 2 – remote work. 3 – work | 2.2 | 0.1 |
| Displacement | 1 – no displacement, 2 – public transportation, 3 – car | 2.4 | 0.1 |

5–point Likert scale from 1 to 5, 1 – minimal, 3 – medium, 5 – maximum; sem – standard error of the mean.

**Appendix 7—table 2.** Linear Mixed-Effects Model results fitting the average Belief Updates (UPD) in participants tested outside (n=58) and during (n=65) the pandemic.

UPD ~1 + context +valence + EE+confidence + age+gender + education +design + valence*context + (1 | subject) + (1+valence | subject) + (1+EE | subject)

**Model fit statistics:**

| AIC | BIC | LogLikelihood | Deviance |
|---|---|---|---|
| 1851.1 | 1913.9 | –907.54 | 1815.1 |

**Fixed effects coefficients (95% CIs):**

*Appendix 7—table 2 Continued on next page*

*Appendix 7—table 2 Continued*

| Name | Estimate | SE | tStat | DF | pValue | 95% CIs Lower | Upper |
|------|----------|-----|-------|-----|--------|---------------|-------|
| Intercept | –2.1975 | 5.033 | –0.4366 | 232 | 0.6628 | –12.114 | 7.7188 |
| valence | 3.2418 | 1.2297 | 2.6363 | 232 | 0.00895 | 0.81898 | 5.6646 |
| context | –0.76328 | 1.9987 | –0.3819 | 232 | 0.7029 | –4.7013 | 3.1747 |
| EE | 0.4187 | 0.11434 | 3.662 | 232 | 0.00031 | 0.19343 | 0.64397 |
| confidence | 0.02234 | 0.03859 | 0.5789 | 232 | 0.56323 | –0.0537 | 0.09836 |
| age | –0.00690 | 0.03181 | –0.2169 | 232 | 0.82845 | –0.0696 | 0.05577 |
| gender | –0.62168 | 1.0909 | –0.5698 | 232 | 0.56932 | –2.7711 | 1.5277 |
| education | –0.29937 | 0.29786 | –1.005 | 232 | 0.31592 | –0.8862 | 0.2875 |
| design | 1.797 | 1.9642 | 0.9149 | 232 | 0.36121 | –2.0729 | 5.6669 |
| valence:context | –5.5395 | 1.6879 | –3.2819 | 232 | 0.00119 | –8.8652 | –2.2139 |

**Random effects covariance parameters (95% CIs):**

Group: subject (121 Levels)

| Name1 | Name2 | Type | Estimate | 95% CIs Lower | Upper |
|-------|-------|------|----------|---------------|-------|
| Intercept | Intercept | std | 2.0778 | NaN | NaN |

Group: subject (121 Levels)

| Name1 | Name2 | Type | Estimate | 95% CIs Lower | Upper |
|-------|-------|------|----------|---------------|-------|
| Intercept | Intercept | std | 3.1204 | NaN | NaN |
| valence | Intercept | corr | –0.38448 | NaN | NaN |
| valence | valence | std | 8.3125 | NaN | NaN |

Group: subject (121 Levels)

| Name1 | Name2 | Type | Estimate | 95% CIs Lower | Upper |
|-------|-------|------|----------|---------------|-------|
| Intercept | Intercept | std | 12.117 | 6.4527 | 22.755 |
| EE | Intercept | corr | –0.99998 | NaN | NaN |
| EE | EE | std | 0.53486 | 0.31495 | 0.90834 |

Group: Error

| Name | Estimate | 95% CIs Lower | Upper |
|------|----------|---------------|-------|
| Res Std | 5.0881 | NaN | NaN |

**Appendix 7—table 3.** Linear Mixed-Effects Model results fitting the average Belief Updates (UPD) in participants tested outside (n=58) and during (n=65) the pandemic, corrected for distance defined by the difference between the estimate for oneself (E1) and for others (eBR).

UPD ~1 + context +valence + EE+confidence + distance +age + gender +education + design +valence*context + (1 | subject) + (1+valence | subject) + (1+EE | subject)

**Model fit statistics:**

| AIC | BIC | LogLikelihood | Deviance |
|-----|-----|---------------|----------|
| 1852.8 | 1919.1 | −907.42 | 1814.8 |

**Fixed effects coefficients (95% CIs):**

| Name | Estimate | SE | tStat | DF | pValue | 95% CIs Lower | Upper |
|------|----------|-----|-------|-----|--------|-------|-------|
| Intercept | −1.6107 | 5.1877 | −0.3105 | 231 | 0.7565 | −11.832 | 8.6105 |
| valence | 3.2539 | 1.2307 | 2.6439 | 231 | 0.0088 | 0.82908 | 5.6788 |
| context | −0.7479 | 1.9932 | −0.3752 | 231 | 0.7078 | −4.6751 | 3.1793 |
| EE | 0.4324 | 0.11745 | 3.6817 | 231 | 0.0003 | 0.201 | 0.66382 |
| confidence | 0.0191 | 0.0391 | 0.4892 | 231 | 0.6252 | −0.0578 | 0.0961 |
| distance | −0.0619 | 0.1262 | −0.500 | 231 | 0.6239 | −0.3105 | 0.1866 |
| age | −0.0075 | 0.0317 | −0.2371 | 231 | 0.8128 | −0.0700 | 0.0550 |
| gender | −0.6504 | 1.0889 | −0.5973 | 231 | 0.5509 | −2.7959 | 1.4951 |
| education | −0.3289 | 0.3053 | −1.0775 | 231 | 0.2824 | −0.9304 | 0.2725 |
| design | 1.8887 | 1.9654 | 0.9610 | 231 | 0.3376 | −1.9837 | 5.7611 |
| valence:context | −5.5767 | 1.6903 | −3.2992 | 231 | 0.0011 | −8.9071 | −2.2463 |

**Random effects covariance parameters (95% CIs):**

Group: subject (121 Levels)

| Name1 | Name2 | Type | Estimate | 95% CIs Lower | Upper |
|-------|-------|------|----------|-------|-------|
| Intercept | Intercept | std | 3.3422 | NaN | NaN |
| valence | Intercept | corr | −0.3646 | NaN | NaN |
| valence | valence | std | 8.3923 | NaN | NaN |

Group: subject (121 Levels)

| Name1 | Name2 | Type | Estimate | 95% CIs Lower | Upper |
|-------|-------|------|----------|-------|-------|
| Intercept | Intercept | std | 12.344 | NaN | NaN |
| EE | Intercept | corr | −0.9994 | NaN | NaN |
| EE | EE | std | 0.5443 | 0.3238 | 0.9149 |

Group: Error

*Appendix 7—table 3 Continued on next page*

*Appendix 7—table 3 Continued*

| Name | Estimate | 95% CIs | |
| --- | --- | --- | --- |
| | | Lower | Upper |
| Res Std | 4.8289 | NaN | NaN |

**Appendix 7—table 4.** Linear Mixed-Effects Model results fitting the average Belief Updates (UPD) in participants tested before the COVID-19 outbreak in France (October 2019, n=30, baseline), and comparing them to participants tested during the first lockdown in March/April 2020 (n=34, context 1), 1 year later in May 2021 with less strict measures in place (n=31, context 2), and at the lift of the sanitary state of emergency in June 2022 (n=28, context 3).

UPD ~1 + context +valence + EE+confidence + age+gender + education +design + valence*context + (1 | subject) + (1+valence | subject) + (1+EE | subject)

**Model fit statistics:**

| AIC | BIC | LogLikelihood | Deviance |
| --- | --- | --- | --- |
| 1857 | 1933.8 | –906.5 | 1813 |

**Fixed effects coefficients (95% CIs):**

| Name | Estimate | SE | tStat | DF | pValue | 95% CIs | |
| --- | --- | --- | --- | --- | --- | --- | --- |
| | | | | | | Lower | Upper |
| Intercept | –5.041 | 6.1791 | –0.8158 | 228 | 0.41546 | –17.216 | 7.1345 |
| valence | 4.2371 | 1.6732 | 2.5324 | 228 | 0.012 | 0.94033 | 7.534 |
| EE | 0.42432 | 0.11572 | 3.6668 | 228 | 0.00031 | 0.1963 | 0.65233 |
| confidence | 0.02225 | 0.0386 | 0.57687 | 228 | 0.5646 | –0.0538 | 0.098252 |
| age | –0.0062 | 0.0317 | –0.1955 | 228 | 0.8452 | –0.0686 | 0.0562 |
| gender | –0.4786 | 1.1083 | –0.4319 | 228 | 0.6663 | –2.6625 | 1.7052 |
| education | –0.2686 | 0.2992 | –0.8978 | 228 | 0.3703 | –0.8582 | 0.3210 |
| design | 2.7278 | 2.2643 | 1.2047 | 228 | 0.2296 | –1.734 | 7.1894 |
| Context 1 | 0.9516 | 2.6961 | 0.3529 | 228 | 0.7245 | –4.361 | 6.2641 |
| Context 2 | 0.5282 | 2.7112 | 0.1948 | 228 | 0.8457 | –4.814 | 5.8703 |
| Context 3 | 1.7166 | 1.8127 | 0.9470 | 228 | 0.3446 | –1.855 | 5.2883 |
| Valence by context 1 | –7.3853 | 2.2942 | –3.2191 | 228 | 0.0015 | –11.906 | –2.8647 |
| Valence by context 2 | –5.5876 | 2.3627 | –2.365 | 228 | 0.0189 | –10.243 | –0.9321 |
| Valence by context 3 | –2.1096 | 2.456 | –0.8590 | 228 | 0.3913 | –6.9489 | 2.7297 |

**Random effects covariance parameters (95% CIs):**

Group: subject (121 Levels)

| Name1 | Name2 | Type | Estimate | 95% CIs | |
| --- | --- | --- | --- | --- | --- |
| | | | | Lower | Upper |
| Intercept | Intercept | std | 1.8256 | NaN | NaN |

*Appendix 7—table 4 Continued on next page*

*Appendix 7—table 4 Continued*

Group: subject (121 Levels)

| Name1 | Name2 | Type | Estimate | 95% CIs | |
| | | | | Lower | Upper |
| --- | --- | --- | --- | --- | --- |
| Intercept | Intercept | std | 2.7322 | NaN | NaN |
| valence | Intercept | corr | –0.4591 | NaN | NaN |
| valence | valence | std | 8.0804 | NaN | NaN |

Group: subject (121 Levels)

| Name1 | Name2 | Type | Estimate | 95% CIs | |
| | | | | Lower | Upper |
| --- | --- | --- | --- | --- | --- |
| Intercept | Intercept | std | 12.429 | 6.8097 | 22.687 |
| EE | Intercept | corr | –0.99996 | NaN | NaN |
| EE | EE | std | 0.54549 | 0.32728 | 0.90917 |

Group: Error

| Name | Estimate | 95% CIs | |
| | | Lower | Upper |
| --- | --- | --- | --- |
| Res Std | 5.6331 | NaN | NaN |

**Appendix 7—table 5.** Linear Mixed-Effects Model results fitting the average Belief Updates (UPD) in participants tested both before and during the pandemic (n=28).

UPD ~1 + context +valence + EE+confidence + age+gender + education +valence*context + (1 | subject) + (1+valence | subject) + (1+EE | subject)

Model fit statistics:

| AIC | BIC | LogLikelihood | Deviance |
| --- | --- | --- | --- |
| 830.29 | 876.5 | –398.14 | 796.29 |

Fixed effects coefficients (95% CIs):

| Name | Estimate | SE | tStat | DF | pValue | 95% CIs | |
| | | | | | | Lower | Upper |
| --- | --- | --- | --- | --- | --- | --- | --- |
| Intercept | 10.333 | 7.5518 | 1.3683 | 103 | 0.1742 | –4.6443 | 25.31 |
| valence | 4.0497 | 1.2219 | 3.3142 | 103 | 0.00127 | 1.6263 | 6.4732 |
| context | –4.4101 | 1.4926 | –2.9546 | 103 | 0.00388 | –7.3704 | –1.4499 |
| EE | 0.08869 | 0.14664 | 0.6046 | 103 | 0.54678 | –0.2022 | 0.37948 |
| confidence | 0.00141 | 0.04768 | 0.02948 | 103 | 0.97654 | –0.0932 | 0.09597 |
| age | –0.05015 | 0.08563 | –0.5856 | 103 | 0.55941 | –0.2199 | 0.11968 |
| gender | 3.5299 | 1.6424 | 2.1492 | 103 | 0.03396 | 0.27253 | 6.7873 |
| education | –0.5649 | 0.42109 | –1.3415 | 103 | 0.1827 | –1.4 | 0.27023 |

*Appendix 7—table 5 Continued on next page*

*Appendix 7—table 5 Continued*

| valence:context | –7.6601 | 1.4923 | –5.1332 | 103 | 1.35e-06 | –10.62 | –4.7005 |

**Random effects covariance parameters (95% CIs):**

**Group: subject (28 Levels)**

| | | | | 95% CIs | |
| --- | --- | --- | --- | --- | --- |
| Name1 | Name2 | Type | Estimate | Lower | Upper |
| Intercept | Intercept | std | 2.2282e-07 | NaN | NaN |

**Group: subject (121 Levels)**

| | | | | 95% CIs | |
| --- | --- | --- | --- | --- | --- |
| Name1 | Name2 | Type | Estimate | Lower | Upper |
| Intercept | Intercept | std | 1.4291 | 0.80997 | 2.5215 |
| valence | Intercept | corr | -1 | NaN | NaN |
| valence | valence | std | 3.3249 | 1.9554 | 5.6537 |

**Group: subject (28 Levels)**

| | | | | 95% CIs | |
| --- | --- | --- | --- | --- | --- |
| Name1 | Name2 | Type | Estimate | Lower | Upper |
| Intercept | Intercept | std | 8.274e-07 | NaN | NaN |
| EE | Intercept | corr | –0.99994 | NaN | NaN |
| EE | EE | std | 3.0154e-08 | NaN | NaN |

**Group: Error**

| | | 95% CIs | |
| --- | --- | --- | --- |
| Name | Estimate | Lower | Upper |
| Res Std | 7.8366 | 6.7914 | 9.0426 |

**Appendix 7—table 6.** Linear Mixed-Effects Model results fitting the average belief updates (UPD) in participants tested outside (n=58) and during (n=65) the pandemic, corrected for distance, and with estimation errors (EE) calculated based on the estimate for someone else (eBR).

UPD ~1 + context +valence + EE+confidence + distance +age + gender +education + design +valence*context + (1 | subject) + (1+valence | subject) + (1+EE | subject)

**Model fit statistics:**

| AIC | BIC | LogLikelihood | Deviance |
| --- | --- | --- | --- |
| 1822.9 | 1889.2 | –892.44 | 1784.9 |

**Fixed effects coefficients (95% CIs):**

| | | | | | | 95% CIs | |
| --- | --- | --- | --- | --- | --- | --- | --- |
| Name | Estimate | SE | tStat | DF | pValue | Lower | Upper |

*Appendix 7—table 6 Continued on next page*

*Appendix 7—table 6 Continued*

| | | | | | | | |
|---|---|---|---|---|---|---|---|
| Intercept | –8.2881 | 5.0873 | –1.6292 | 231 | 0.1046 | –18.311 | 1.7353 |
| valence | 2.4409 | 1.2913 | 1.8902 | 231 | 0.0600 | –0.1034 | 4.9852 |
| context | –0.7860 | 1.7408 | –0.4515 | 231 | 0.6520 | –4.2159 | 2.6438 |
| EE | 0.5057 | 0.1220 | 4.1459 | 231 | 4.8e-05 | 0.2654 | 0.7460 |
| confidence | 0.0948 | 0.0349 | 2.7202 | 231 | 0.0070 | 0.0261 | 0.1635 |
| distance | –0.1582 | 0.1054 | –1.5013 | 231 | 0.1347 | –0.3659 | 0.0494 |
| age | –0.0025 | 0.0274 | –0.0916 | 231 | 0.9271 | –0.0566 | 0.0515 |
| gender | 0.6055 | 0.9504 | 0.6371 | 231 | 0.5247 | –1.2671 | 2.4781 |
| education | –0.2419 | 0.2640 | –0.9160 | 231 | 0.3606 | –0.7621 | 0.2784 |
| design | 0.9142 | 1.7301 | 0.5284 | 231 | 0.5977 | –2.4945 | 4.3229 |
| valence:context | –5.0968 | 1.7627 | –2.8914 | 231 | 0.0042 | –8.5698 | –1.6237 |

**Random effects covariance parameters (95% CIs):**

Group: subject (121 Levels)

| | | | | 95% CIs | |
|---|---|---|---|---|---|
| Name1 | Name2 | Type | Estimate | Lower | Upper |
| Intercept | Intercept | std | 1.5868 | NaN | NaN |

Group: subject (121 Levels)

| | | | | 95% CIs | |
|---|---|---|---|---|---|
| Name1 | Name2 | Type | Estimate | Lower | Upper |
| Intercept | Intercept | std | 2.9714 | NaN | NaN |
| valence | Intercept | corr | 0.2994 | NaN | NaN |
| valence | valence | std | 9.2418 | NaN | NaN |

Group: subject (121 Levels)

| | | | | 95% CIs | |
|---|---|---|---|---|---|
| Name1 | Name2 | Type | Estimate | Lower | Upper |
| Intercept | Intercept | std | 1.1447 | NaN | NaN |
| EE | Intercept | corr | –0.9956 | NaN | NaN |
| EE | EE | std | 0.1645 | NaN | NaN |

Group: Error

| | | 95% CIs | |
|---|---|---|---|
| Name | Estimate | Lower | Upper |
| Res Std | 3.9480 | NaN | NaN |

*Appendix 7—table 7 Continued on next page*

**Appendix 7—table 7.** Linear Mixed-Effects Model results fitting the average confidence ratings in participants tested outside (n=58) and during (n=65) the pandemic.

confidence ~1 + context +valence + EE+age + gender +education + design +valence*context + (1 | subject)

**Model fit statistics:**

| AIC | BIC | LogLikelihood | Deviance |
|-----|-----|---------------|----------|
| 1908.6 | 1946.9 | –943.28 | 1886.6 |

**Fixed effects coefficients**

| Name | Estimate | SE | tStat | DF | pValue | 95% CIs Lower | Upper |
|------|----------|-----|-------|-----|--------|-------|-------|
| Intercept | 61.051 | 9.1552 | 6.6685 | 233 | 1.86e-10 | 43.013 | 79.088 |
| valence | –0.4795 | 0.7288 | –0.6580 | 233 | 0.5112 | –1.9154 | 0.9563 |
| context | 14.105 | 4.524 | 3.1177 | 233 | 0.0021 | 5.1915 | 23.018 |
| EE | –0.2398 | 0.1177 | –2.0376 | 233 | 0.0427 | –0.4717 | –0.0079 |
| age | 0.0451 | 0.0747 | 0.6037 | 233 | 0.5466 | –0.1021 | 0.1923 |
| gender | –2.415 | 2.5148 | –0.9603 | 233 | 0.3379 | –7.3697 | 2.5397 |
| education | 0.0563 | 0.6898 | 0.0816 | 233 | 0.935 | –1.3027 | 1.4153 |
| design | 3.8568 | 4.6503 | 0.8294 | 233 | 0.4077 | –5.3052 | 13.019 |
| valence:context | 0.0969 | 1.0256 | 0.0944 | 233 | 0.9248 | –1.9237 | 2.1174 |

**Random effects covariance parameters**

Group: subject (121 Levels)

| Name1 | Name2 | Type | Estimate | 95% CIs Lower | Upper |
|-------|-------|------|----------|-------|-------|
| Intercept | Intercept | std | 11.892 | 10.188 | 13.88 |

Group: Error

| Name | Estimate | 95% CIs Lower | Upper |
|------|----------|-------|-------|
| Res Std | 7.6935 | 6.7828 | 8.7265 |

**Appendix 7—table 8.** Linear Mixed-Effects Model results fitting the average absolute Estimation Error (EE) in participants tested outside (n=58) and during (n=65) the pandemic.

EE ~1 + context +valence + confidence +age + gender +education + design +valence*context + (1 | subject)

**Model fit statistics:**

| AIC | BIC | LogLikelihood | Deviance |
|-----|-----|---------------|----------|
| 1569 | 1607.3 | –773.48 | 1547 |

**Fixed effects coefficients (95% CIs):**

*Appendix 7—table 8 Continued on next page*

*Appendix 7—table 8 Continued*

| Name | Estimate | SE | tStat | DF | pValue | 95% CIs Lower | Upper |
|---|---|---|---|---|---|---|---|
| Intercept | 24.602 | 3.5887 | 6.8553 | 233 | 6.32e-11 | 17.531 | 31.672 |
| valence | −0.4550 | 0.4902 | −0.9282 | 233 | 0.3543 | −1.4209 | 0.5108 |
| context | 3.686 | 1.6893 | 2.1819 | 233 | 0.0301 | 0.3577 | 7.0143 |
| confidence | −0.0496 | 0.0289 | −1.7123 | 233 | 0.0882 | −0.1067 | 0.0075 |
| age | 0.0403 | 0.0272 | 1.4801 | 233 | 0.1402 | −0.0133 | 0.0939 |
| gender | −1.073 | 0.9181 | −1.1688 | 233 | 0.2437 | −2.8818 | 0.7358 |
| education | −0.3296 | 0.2511 | −1.3128 | 233 | 0.1906 | −0.8243 | 0.1651 |
| design | 1.7037 | 1.6971 | 1.0039 | 233 | 0.3165 | −1.64 | 5.0473 |
| valence:context | 2.1942 | 0.6688 | 3.2808 | 233 | 0.0012 | 0.8765 | 3.5119 |

**Random effects covariance parameters**

Group: subject (121 Levels)

| Name1 | Name2 | Type | Estimate | 95% CIs Lower | Upper |
|---|---|---|---|---|---|
| Intercept | Intercept | std | 3.046 | 2.1255 | 4.3652 |

Group: Error

| Name | Estimate | 95% CIs Lower | Upper |
|---|---|---|---|
| Res Std | 5.1867 | 4.5717 | 5.8845 |

**Appendix 7—table 9.** Linear Mixed-Effects Model results fitting the average Learning Rates from the RL-like model in participants tested outside (n=58) and during (n=65) the pandemic.

LR ~1 + context +valence + age+gender + education +design + valence*context + (1 | subject)

**Model fit statistics:**

| AIC | BIC | LogLikelihood | Deviance |
|---|---|---|---|
| −54.291 | −19.319 | 37.146 | −74.291 |

**Fixed effects coefficients**

| Name | Estimate | SE | tStat | DF | pValue | 95% CIs Lower | Upper |
|---|---|---|---|---|---|---|---|
| Intercept | 0.5189 | 0.1546 | 3.3566 | 236 | 0.0009 | 0.2144 | 0.8235 |
| valence | 0.0856 | 0.0120 | 7.1351 | 236 | 1.18e-11 | 0.0620 | 0.1092 |
| context | −0.0492 | 0.0797 | −0.6172 | 236 | 0.5377 | −0.2062 | 0.1078 |
| age | −0.0005 | 0.0014 | −0.3404 | 236 | 0.7339 | −0.0031 | 0.0022 |
| gender | −0.0287 | 0.0457 | −0.6267 | 236 | 0.5315 | −0.1187 | 0.0614 |

*Appendix 7—table 9 Continued on next page*

*Appendix 7—table 9 Continued*

| | | | | | | | | |
|---|---|---|---|---|---|---|---|---|
| education | −0.0227 | 0.0126 | −1.8091 | 236 | 0.0717 | | −0.0474 | 0.0020 |
| design | 0.0269 | 0.0822 | 0.3272 | 236 | 0.7438 | | −0.1351 | 0.1889 |
| valence:context | −0.0347 | 0.0164 | −2.1126 | 236 | 0.0357 | | −0.0671 | −0.0023 |

**Random effects covariance parameters**

**Group: subject (122 Levels)**

| | | | | 95% CIs | |
|---|---|---|---|---|---|
| Name1 | Name2 | Type | Estimate | Lower | Upper |
| Intercept | Intercept | std | 0.22053 | 0.19016 | 0.25575 |

**Group: Error**

| | 95% CIs | |
|---|---|---|
| Name | Estimate | Lower | Upper |
|---|---|---|---|
| Res Std | 0.12808 | 0.11297 | 0.1452 |

**Appendix 7—table 10.** Linear Mixed-Effects Model results fitting the average Learning Rates for RL-like model in participants tested both before and during the pandemic (n=28).

LR ~1 + context +valence + age+gender + education +valence*context + (1 | subject)

**Model fit statistics:**

| AIC | BIC | LogLikelihood | Deviance |
|---|---|---|---|
| −15.954 | 8.5125 | 16.977 | −33.954 |

**Fixed effects coefficients**

| | | | | | | 95% CIs | |
|---|---|---|---|---|---|---|---|
| Name | Estimate | SE | tStat | DF | pValue | Lower | Upper |
| Intercept | 0.3529 | 0.1869 | 1.8887 | 105 | 0.0617 | −0.0176 | 0.7234 |
| valence | 0.0752 | 0.0234 | 3.2175 | 105 | 0.0017 | 0.0288 | 0.1215 |
| context | −0.1024 | 0.0330 | −3.1011 | 105 | 0.0025 | −0.1679 | −0.0369 |
| age | 0.0010 | 0.0035 | 0.2876 | 105 | 0.7742 | −0.0060 | 0.0080 |
| gender | 0.1858 | 0.0704 | 2.6385 | 105 | 0.0096 | 0.0462 | 0.3254 |
| education | −0.0216 | 0.0178 | −1.2130 | 105 | 0.2279 | −0.0570 | 0.0137 |
| valence:context | −0.0616 | 0.0330 | −1.8662 | 105 | 0.0648 | −0.1271 | 0.0039 |

**Random effects covariance parameters**

**Group: subject (28 Levels)**

| | | | | 95% CIs | |
|---|---|---|---|---|---|
| Name1 | Name2 | Type | Estimate | Lower | Upper |
| Intercept | Intercept | std | 0.15167 | 0.10662 | 0.21576 |

*Appendix 7—table 10 Continued on next page*

*Appendix 7—table 10 Continued*

Group: Error

| | | 95% CIs | |
|---|---|---|---|
| Name | Estimate | Lower | Upper |
| Res Std | 0.17479 | 0.15026 | 0.20332 |

**Appendix 7—table 11.** Linear Mixed-Effects Model results fitting the average asymmetry in the RL-like model in participants tested outside (n=58) and during (n=65) the pandemic.

asymmetry ~1 + context +age + gender +education + (1 | subject)

Model fit statistics:

| AIC | BIC | LogLikelihood | Deviance |
|---|---|---|---|
| −204.79 | −185.16 | 109.39 | −218.79 |

Fixed effects coefficients

| Name | Estimate | SE | tStat | DF | pValue | 95% CIs Lower | Upper |
|---|---|---|---|---|---|---|---|
| Intercept | 0.0264 | 0.0323 | 0.818 | 117 | 0.415 | −0.0375 | 0.0903 |
| context | −0.04 | 0.0172 | −2.3202 | 117 | 0.0221 | −0.0741 | −0.0058 |
| age | 0.0006 | 0.0005 | 1.1752 | 117 | 0.2423 | −0.0004 | 0.0016 |
| gender | 0.0089 | 0.0045 | 2.0024 | 117 | 0.0475 | 0.0001 | 0.0178 |
| education | −0.0048 | 0.0172 | −0.2815 | 117 | 0.7788 | −0.0388 | 0.0291 |

Random effects covariance parameters

Group: subject (122 Levels)

| Name1 | Name2 | Type | Estimate | 95% CIs Lower | Upper |
|---|---|---|---|---|---|
| Intercept | Intercept | std | 0.063415 | NaN | NaN |

Group: Error

| Name | Estimate | 95% CIs Lower | Upper |
|---|---|---|---|
| Res Std | 0.063415 | NaN | NaN |

**Appendix 7—table 12.** Linear Mixed-Effects Model results fitting the average scaling in the RL-like model in participants tested outside (n=58) and during (n=65) the pandemic.

scaling ~1 + context +age + gender +education + (1 | subject)

Model fit statistics:

| AIC | BIC | LogLikelihood | Deviance |
|---|---|---|---|

*Appendix 7—table 12 Continued on next page*

*Appendix 7—table 12 Continued*

| 10.519 | 30.147 | 1.7406 | −3.4813 |
|--------|--------|--------|---------|

#### Fixed effects coefficients

| Name | Estimate | SE | tStat | DF | pValue | 95% CIs Lower | Upper |
|------|----------|-----|-------|-----|--------|-------|-------|
| Intercept | 0.561 | 0.0858 | 6.5375 | 117 | 1.71e-09 | 0.3911 | 0.731 |
| context | −0.0705 | 0.0458 | −1.5406 | 117 | 0.1261 | −0.1612 | 0.0201 |
| age | −0.0004 | 0.0014 | −0.3263 | 117 | 0.7448 | −0.0031 | 0.0022 |
| gender | −0.0214 | 0.0119 | −1.7993 | 117 | 0.0745 | −0.0449 | 0.0022 |
| education | −0.0297 | 0.0456 | −0.6517 | 117 | 0.5159 | −0.1201 | 0.0606 |

#### Random effects covariance parameters

Group: subject (122 Levels)

| Name1 | Name2 | Type | Estimate | 95% CIs Lower | Upper |
|-------|-------|------|----------|-------|-------|
| Intercept | Intercept | std | 0.16865 | NaN | NaN |

Group: Error

| Name | Estimate | 95% CIs Lower | Upper |
|------|----------|-------|-------|
| Res Std | 0.16865 | NaN | NaN |

**Appendix 7—table 13.** Linear Mixed-Effects Model results fitting the average asymmetry in the RL-like model in participants tested both before and during the pandemic (n=28).

asymmetry ~1 + context +age + gender +education + (1 | subject)

#### Model fit statistics:

| AIC | BIC | LogLikelihood | Deviance |
|-----|-----|---------------|----------|
| −138.71 | −124.53 | 76.354 | −152.71 |

#### Fixed effects coefficients

| Name | Estimate | SE | tStat | DF | pValue | 95% CIs Lower | Upper |
|------|----------|-----|-------|-----|--------|-------|-------|
| Intercept | 0.0471 | 0.0473 | 0.9969 | 51 | 0.3235 | −0.0478 | 0.1420 |
| context | −0.0615 | 0.0165 | −3.7185 | 51 | 0.0005 | −0.0947 | −0.0283 |
| age | 0.0005 | 0.0009 | 0.6121 | 51 | 0.5432 | −0.0012 | 0.0023 |
| gender | −0.0023 | 0.0176 | −0.1285 | 51 | 0.8983 | −0.0376 | 0.0331 |
| education | 0.0019 | 0.0045 | 0.4158 | 51 | 0.6793 | −0.0071 | 0.0108 |

*Appendix 7—table 13 Continued on next page*

*Appendix 7—table 13 Continued*

**Random effects covariance parameters**

Group: subject (28 Levels)

| Name1 | Name2 | Type | Estimate | 95% CIs | |
| | | | | Lower | Upper |
|---|---|---|---|---|---|
| Intercept | Intercept | std | 9.1905e-09 | NaN | NaN |

Group: Error

| Name | Estimate | 95% CIs | |
| | | Lower | Upper |
|---|---|---|---|
| Res Std | 0.06189 | 0.051427 | 0.074482 |

**Appendix 7—table 14.** Linear Mixed-Effects Model results fitting the average scaling in the RL-like model in participants tested both before and during the pandemic (n=28).

scaling ~1 + context +age + gender +education + (1 | subject)

**Model fit statistics:**

| AIC | BIC | LogLikelihood | Deviance |
|---|---|---|---|
| 3.4809 | 17.658 | 5.2595 | −10.519 |

**Fixed effects coefficients**

| Name | Estimate | SE | tStat | DF | pValue | 95% CIs | |
| | | | | | | Lower | Upper |
|---|---|---|---|---|---|---|---|
| Intercept | 0.3529 | 0.1880 | 1.8776 | 51 | 0.0662 | −0.0244 | 0.7302 |
| context | −0.1024 | 0.0524 | −1.9554 | 51 | 0.0560 | −0.2076 | 0.0027 |
| age | 0.0010 | 0.0035 | 0.2876 | 51 | 0.7748 | −0.0061 | 0.0081 |
| gender | 0.1858 | 0.0704 | 2.6385 | 51 | 0.0110 | 0.0444 | 0.3272 |
| education | −0.0216 | 0.0178 | −1.2130 | 51 | 0.2307 | −0.0575 | 0.0142 |

**Random effects covariance parameters**

Group: subject (28 Levels)

| Name1 | Name2 | Type | Estimate | 95% CIs | |
| | | | | Lower | Upper |
|---|---|---|---|---|---|
| Intercept | Intercept | std | 0.10692 | 0.046689 | 0.24485 |

Group: Error

| Name | Estimate | 95% CIs | |
| | | Lower | Upper |
|---|---|---|---|
| Res Std | 0.19601 | 0.15084 | 0.2547 |

**Appendix 7—table 15.** Sociodemographical data (N=123).

|  | N | gender | age | education level |
|---|---|---|---|---|
| Outside pandemic | 58 | 32 females | 33.84±1.68 | 4.54±0.29 |
| During pandemic | 65 | 45 females | 42.32±2.35 | 4.59±0.19 |
| Group tested before and during | 28 | 18 females | 34.14±2.08 | 5.00±0.41 |

Note: education is reported as years of higher education (university level).

**Appendix 7—table 16.** Linear Mixed-Effects Model results fitting the average number of paradoxical trials in participants tested outside (n=58) and during (n=65) the pandemic.

Nb of trials ~1 + valence +context + age+gender + education +design + context*valence + (1 | subject)

Model fit statistics:

| AIC | BIC | LogLikelihood | Deviance |
|---|---|---|---|
| 1623.5 | 1644.6 | –805.76 | 1611.5 |

Fixed effects coefficients

| Name | Estimate | SE | tStat | DF | pValue | 95% CIs Lower | Upper |
|---|---|---|---|---|---|---|---|
| Intercept | 19.427 | 3.059 | 6.3508 | 234 | 1.10e-09 | 13.4 | 25.454 |
| valence | –1.9732 | 0.6083 | –3.2436 | 234 | 0.0014 | –3.1717 | –0.7747 |
| context | –0.1904 | 1.57 | –0.1213 | 234 | 0.9036 | –3.2836 | 2.9028 |
| age | 0.0010 | 0.0260 | 0.0402 | 234 | 0.9680 | –0.0501 | 0.0522 |
| gender | –0.1403 | 0.8746 | –0.1604 | 234 | 0.8727 | –1.8633 | 1.5828 |
| education | 0.0022 | 0.2397 | 0.0092 | 234 | 0.9926 | –0.4701 | 0.4745 |
| design | 0.1215 | 1.6177 | 0.0751 | 234 | 0.9402 | –3.0656 | 3.3085 |
| context:valence | –2.3576 | 0.8300 | –2.8404 | 234 | 0.0049 | –3.9928 | –0.7223 |

Random effects covariance parameters

Group: subject (123 Levels)

| Name1 | Name2 | Type | Estimate | 95% CIs Lower | Upper |
|---|---|---|---|---|---|
| Intercept | Intercept | std | 2.8591e-15 | NaN | NaN |

Group: Error

| Name | Estimate | 95% CIs Lower | Upper |
|---|---|---|---|
| Res Std | 6.4381 | 5.8893 | 7.038 |

*Appendix 7—table 17 Continued on next page*

**Appendix 7—table 17.** Linear Mixed-Effects Model results fitting the average number of under- and overshooting in participants tested outside (n=58) and during (n=65) the pandemic.

Nb of shoot ~1 + type+context + EE+age + gender +education + design +context*type + (1 | subject)

Model fit statistics:

| AIC | BIC | LogLikelihood | Deviance |
|---|---|---|---|
| 1623.5 | 1644.6 | –805.76 | 1611.5 |

Fixed effects coefficients

| | | | | | | 95% CIs | |
|---|---|---|---|---|---|---|---|
| Name | Estimate | SE | tStat | DF | pValue | Lower | Upper |
| Intercept | 3.5229 | 2.0186 | 1.7452 | 233 | 0.0823 | –0.4542 | 7.5001 |
| type | –1.0751 | 0.3611 | –2.9769 | 233 | 0.0032 | –1.7866 | –0.3636 |
| context | 0.2141 | 0.9390 | 0.2280 | 233 | 0.8199 | –1.6359 | 2.064 |
| EE | –0.0288 | 0.0406 | –0.7102 | 233 | 0.4783 | –0.1088 | 0.0511 |
| age | 0.0079 | 0.0155 | 0.5099 | 233 | 0.6106 | –0.0226 | 0.0384 |
| gender | –0.4039 | 0.5200 | –0.7766 | 233 | 0.4382 | –1.4284 | 0.6207 |
| education | –0.4182 | 0.1428 | –2.9286 | 233 | 0.0037 | –0.6995 | –0.1369 |
| design | 1.4082 | 0.9612 | 1.4651 | 233 | 0.1443 | –0.4855 | 3.3019 |
| context:type | 1.6648 | 0.5003 | 3.3277 | 233 | 0.0010 | 0.6792 | 2.6505 |

Random effects covariance parameters

Group: subject (123 Levels)

| | | | | 95% CIs | |
|---|---|---|---|---|---|
| Name1 | Name2 | Type | Estimate | Lower | Upper |
| Intercept | Intercept | std | 2.2037e-06 | NaN | NaN |

Group: Error

| | | 95% CIs | |
|---|---|---|---|
| Name | Estimate | Lower | Upper |
| Res Std | 3.8174 | 3.492 | 4.1731 |

**Appendix 7—table 18.** Linear Mixed-Effects Model results fitting initial beliefs about the likelihood of adverse future life events for oneself (E1) and for others (eBR) in participants tested outside (n=58) and during (n=65) the pandemic.
Note the perspective regressor (coded 0 for E1 and 1 for eBR) tested if and how beliefs differed when assessed for oneself than for others.

Estimate ~1 + context +perspective + EE+confidence + age+gender + education +design + context*perspective + (1 | subject)

Model fit statistics:

| AIC | BIC | LogLikelihood | Deviance |
|---|---|---|---|

*Appendix 7—table 18 Continued on next page*

*Appendix 7—table 18 Continued*

| 1630.4 | 1672.3 | −803.22 | 1606.4 |
|---|---|---|---|

**Fixed effects coefficients (95% CIs):**

| Name | Estimate | SE | tStat | DF | pValue | 95% CIs Lower | Upper |
|---|---|---|---|---|---|---|---|
| Intercept | 31.472 | 6.6809 | 4.7108 | 232 | 4.25e-06 | 18.309 | 44.635 |
| perspective | 3.0194 | 0.8564 | 3.5257 | 232 | 0.0005 | 1.3321 | 4.7067 |
| context | −1.9127 | 2.9993 | −0.6377 | 232 | 0.5243 | −7.8221 | 3.9966 |
| EE | 0.3146 | 0.1261 | 2.4955 | 232 | 0.0133 | 0.0662 | 0.5629 |
| confidence | 0.0598 | 0.0350 | 1.7069 | 232 | 0.0892 | −0.0092 | 0.1287 |
| age | −0.0808 | 0.0481 | −1.6793 | 232 | 0.09444 | −0.1756 | 0.0140 |
| gender | 0.1048 | 1.6285 | 0.0643 | 232 | 0.9488 | −3.1037 | 3.3133 |
| education | −0.6857 | 0.4443 | −1.5433 | 232 | 0.1241 | −1.5612 | 0.1897 |
| design | 0.2792 | 2.9999 | 0.0931 | 232 | 0.9259 | −5.6313 | 6.1897 |
| context: perspective | 0.0678 | 0.9811 | 0.0691 | 232 | 0.9450 | −1.8652 | 2.0007 |

**Random effects covariance parameters (95% CIs):**

Group: subject (121 Levels)

| Name1 | Name2 | Type | Estimate | 95% CIs Lower | Upper |
|---|---|---|---|---|---|
| Intercept | Intercept | std | 8.0025 | 6.9443 | 9.2219 |

Group: Error

| Name | Estimate | 95% CIs Lower | Upper |
|---|---|---|---|
| Res Std | 3.7506 | 3.3033 | 4.2584 |

