## [Editor Report · eLife Assessment]

This **important** study addresses the question of how large-scale events such as the COVID-19 pandemic can change people's beliefs and their updates. Using a well-validated task, the authors find that belief updating becomes less optimistically biased during COVID-19 compared to before it. In this revision, due to the addition of more model-based analyses and power calculations, they have generated **convincing** evidence for their primary claim that the pandemic significantly impacted people's belief updating away from optimistic belief updating. As with many manipulations outside the experimenters' control, it remains unclear which psychological factor impacted by the pandemic drives the group differences, and sample sizes are, by necessity, on the smaller side as data cannot readily be acquired. However, the authors are commended for doing power analyses, showing their sensitivity, and recognizing the limitations of their study.

---

## [Referee Report · Reviewer #1 (Public review)]

This manuscript uses a well-validated behavioural estimation task to investigate the degree to which optimistic belief updating was attenuated during the 2020 global pandemic. Online participants estimated how likely different negative life events were to happen to them in the future and were given statistics about these events. Belief updating (measured as the degree to which estimations changed after viewing the statistics) was less optimistically biased during the pandemic (compared to outside of it). This resulted from reduced updating from "good news" (better than expected information). Computational models were used to try to unpack how statistics were integrated and used to revise beliefs. Two families of models were compared - an RL set of models where "estimation errors" (analogous to prediction errors in classic RL models) predict belief change and a Bayesian set of models where an implied likelihood ratio was calculated (derived from participants estimations of their own risk and estimation of the base rate risk) and used to predict belief change. The authors found evidence that the former set of models accounted for updating better outside of the pandemic, but the latter accounted for updating during the pandemic. In addition, the RL model provides evidence that learning was asymmetrically positively biased outside of the pandemic but symmetric during it (as a result of reduced learning rates from good news estimation errors).

Strengths

Understanding whether biases in learning are fixed modes of information processing or flexible and adapt in response to environmental shocks (like a global pandemic or economic recession) is an important area of research relevant to a wide range of fields, including cognitive psychology, behavioural economics, and computational psychiatry. The study uses a well-validated task, and the authors conduct a power analysis to show that the sample sizes are appropriate. Furthermore, the authors test that their results hold in both a between-group analysis (the focus of the main paper) and a within-group analysis (mainly in the supplemental).

The finding that optimistic biases are reduced in response to acute stress, perceived threat, and depression has been shown before using this task both in the lab (social stress manipulation), in the real world (firefighters on duty), and clinical groups (patients with depression). However, the work does extend these findings here in important ways:

(1) Examining the effect of a new real-world adverse event (the pandemic).

(2) The reduction in optimistic updating here arises due to reduced updating from positive information (previously, in the case of environmental threat, this reduction mainly arose from increased sensitivity to negative information).

(3) Leveraging new RL-inspired computational approaches, demonstrating that the bias - and its attenuation - can be captured using trial-by-trial computational modelling with separate learning rates for positive and negative estimation errors.

The authors now take great care to caveat that the findings cannot directly attribute the observed lack of optimistically biased belief updating during lockdown to psychological causes such as heightened anxiety and stress.

The authors have added model recovery results. Whilst there are some cases within a family (RL or Bayesian) of models where they can be confused (e.g., Bayesian model 10-the winning model during the pandemic-sometimes gets confused with Bayesian model 9), there is no confusion between families of models (RL models don't get confused with Bayesian models and vice versa), which is reassuring.

Weaknesses

The authors now conduct model recovery (SI Figure 5) and show how the behaviour of the two best-fitting models (Rational Bayesian model and optimistically biased RL-like model) approximates the actual data observed by showing them alongside each other (Figure 1b). It seems from Figure 1b that the 2 models predict similar behaviour for bad news but diverge for good news, with the optimistically biased RL-like model predicting greater updates than the rational Bayesian model. However, it is difficult to tell from the figure (partly because of the y-axis scale) how much of a divergence this is and how distinctive a pattern relative to the other models. I think the interpretation could be improved further by a clearer sense of the behavioural signatures of each model, enabling them to be reliably teased apart from one another in the model recovery.

---

## [Referee Report · Reviewer #2 (Public review)]

The authors investigated how experiencing the COVID-19 pandemic affected optimism bias in updating beliefs about the future. They ran a between-subjects design testing participants on cognitive tasks before, during and after the lift of the sanitary state of emergency during the pandemic. The authors show that optimism bias varied depending on the context in which it was tested. Namely, it disappeared during COVID-19 and it re-emerged at the time of lift of sanitary emergency measures. Via advanced computational modelling they are able to thoroughly characterise the nature of such alterations, pinpointing specific mechanisms underlying the lack of optimistic bias during the pandemic.

Strengths pertain to the comprehensive assessment of the results via computational modelling, and from a theoretical point of view, the notion that environmental factors can affect cognition. Power analysis was conducted to ensure that the study was powered to observe the effect of interest despite the relatively small sample size.

As the authors also noted, a major impediment to the interpreting the findings pertains to the lack of additional measures. While information on, for example, risk perception or need for social interaction were collected from participants during the pandemic, the fact that these could not be included in the analysis hindered the interpretation of findings. While the interpretation of the findings remains challenging, this work offers an example of the influence of real-life conditions on the belief-updating process.

---

## [Author Response]

The following is the authors’ response to the original reviews

**Reviewer #1:**
Summary:This manuscript uses a well-validated behavioral estimation task to investigate how optimistic belief updating was attenuated during the 2020 global pandemic. Online participants recruited during and outside of the pandemic estimated how likely different negative life events were to happen to them in the future and were given statistics about these events happening. Belief updating (measured as the degree to which estimations changed after viewing the statistics) was less optimistically biased during the pandemic (compared to outside of it). This resulted from reduced updating from "good news" (better than expected information). Computational models were used to try to unpack how statistics were integrated and used to revise beliefs. Two families of models were compared - an RL set of models where "estimation errors" (analogous to prediction errors in classic RL models) predict belief change and a Bayesian set of models where an implied likelihood ratio was calculated (derived from participants estimations of their own risk and estimation of the base rate risk) and used to predict belief change. The authors found evidence that the former set of models accounted for updating better outside of the pandemic, but the latter accounted for updating during the pandemic. In addition, the RL model provides evidence that learning was asymmetrically positively biased outside of the pandemic but symmetric during it (as a result of reduced learning rates from good news estimation errors).Strengths:Understanding whether biases in learning are fixed modes of information processing or flexible and adapt in response to environmental shocks (like a global pandemic or economic recession) is an important area of research relevant to a wide range of fields, including cognitive psychology, behavioral economics, and computational psychiatry. The study uses a well-validated task, and the authors conduct a power analysis to show that the sample sizes are appropriate. Furthermore, the authors test that their results hold in both a between-group analysis (the focus of the main paper) and a within-group analysis (mainly in the supplemental).The finding that optimistic biases are reduced in response to acute stress, perceived threat, and depression has been shown before using this task both in the lab (social stress manipulation), in the real world (firefighters on duty), and clinical groups (patients with depression). However, the work does extend these findings here in important ways:(1) Examining the effect of a new real-world adverse event (the pandemic).(2) The reduction in optimistic updating here arises due to reduced updating from positive information (previously, in the case of environmental threat, this reduction mainly arose from increased sensitivity to negative information).(3) Leveraging new RL-inspired computational approaches, demonstrating that the bias - and its attenuation - can be captured using trial-by-trial computational modeling with separate learning rates for positive and negative estimation errors.Weaknesses:Some interpretation and analysis (the computational modeling in particular) could be improved.On the interpretation side, while the pandemic was an adverse experience and stressful for many people (including myself), the absence of any measures of stress/threat levels limits the conclusions one can draw. Past work that has used this task to examine belief updating in response to adverse environmental events took physiological (e.g., SCR, cortisol) and/or self-report (questionnaires) measures of mood. In SI Table 1, the authors possibly had some questionnaire measures along these lines, but this might be for the participants tested during the pandemic.

Thank you for this review.

We agree that the lack of physiological and self-report measures of stress, threat, and perceived uncertainty limits the interpretation of findings regarding potential psychological factors. Some self-reported anxiety and perceived risk measures experienced during the lockdowns were collected in a subset of participants (n=40, counting n=21 tested before and during the 1st strict lockdown, and n=19 tested solely during the 1st lockdown). These reports were given retrospectively at the time of release of the 1st lockdown in summer 2020 when the pandemic was still unfolding (SI Table 1).

Exploratory correlations revealed some noteworthy trends. We found that participants who reported to have perceived a bigger risk of death due to contagion were also those who were less optimistically biased when updating their beliefs about adverse future life risks during the first strict COVID-19-related lockdown (Pearson’s r = -0.36, p = 0.02).

Moreover, parameter estimates from the computational models of belief updating showed associations with specific survey responses: The rational Bayesian model’s scaling parameter correlated positively with adherence to distancing measures (r = 0.41, p = 0.01) and negatively with the need for social contact (r = -0.37, p = 0.02). This result indicated that participants who were updating their beliefs faster were more likely to follow preventive guidelines and felt less social craving. Meanwhile, the asymmetry parameter correlated negatively with mask wearing (r = -0.41, p = 0.01), positively with physical contact with close others (r = 0.32, p = 0.04) and satisfaction with social interactions (r = 0.33, p = 0.04). This suggests that participants who displayed some asymmetry in belief updating during the COVID-19 pandemic were less likely to comply with mask-wearing rules and more likely to engage in social interactions.

However, these results did not survive correction for multiple comparisons and the sample size for correlational analyses is in the lower range. The subjective measures of anxiety and fear of contagion did not significantly correlate to the updating bias, or any other variable measured by the belief updating task (e.g. estimation error, updating magnitude).

We now further discuss on page 12 the limitation, which reads:

“We did not collect physiological measures of stress or information about the COVID-19 infection status of participants, which precludes a direct exploration of the immediate effects of experiencing the infection on belief-updating behavior and the potential interaction with anxiety and stress levels. Although subjective ratings of the perceived risk of death from COVID-19 correlated negatively to the beliefs updating bias measured during the pandemic, this result was obtained retrospectively in a subset of participants (SI section 4). We thus cannot directly attribute the observed lack of optimistically biased belief updating during the lockdown to psychological causes such as heightened anxiety and stress. This limitation is noteworthy, as the impact of experiencing the pandemic on belief updating about the future could differ between those who directly experienced infection and those who remained uninfected. It is also important to acknowledge that our study was timely and geographically limited to the context of the COVID-19 outbreak in France. Cultural variations and differences in governmental responses to contain the spread of SARS-CoV-2 may have impacted the optimism biases in belief updating differently.”

On the analysis side, it was unclear what the motivation was for the different sets of models tested. Both families of models test asymmetric vs symmetric learning (which is the main question here) and have similar parameters (scaling and asymmetry parameters) to quantify these different aspects of the learning process. Conceptually, the different behavioral patterns one could expect from the two families of models needed to be clarified.

Thank you for raising this point. We agree that a clearer conceptual distinction between the two model families can help strengthen the interpretation of our findings. We have added the following considerations to the introduction on pages 2–3, which now reads:

“The underlying mechanism of optimistically biased belief updating involves an asymmetry in learning from positive and negative belief-disconfirming information[2,3,4], which can unfold in two ways following Reinforcement learning (RL) or Bayes rule[5].

Conceptually, Reinforcement learning (RL) and Bayesian models of belief updating are complementary but make different assumptions about the hidden process humans may use to adjust their beliefs when faced with information that contradicts them. The RL models assume belief updating is proportional to the estimation error. The key idea of the estimation error expresses the difference between how much someone believes they will experience a future life event and the actual prevalence of the event in the general population. This difference can be positive or negative. A scaling and an asymmetry parameter quantify the propensity to consider the estimation error magnitude and its valence, respectively. These two free parameters form the learning rate, which indicates how fast and biased participants update their beliefs.

In contrast, Bayesian models assume that following Bayes’ rule the posterior, updated belief is a new hypothesis, formed by pondering prior knowledge with new evidence. The prior knowledge consists in information about the prevalence of life events in the general population. The new evidence comprises various alternative hypotheses. It examines how likely a specific event is to occur or not occur for oneself, compared to the likelihood that it will happen or not happen to others. This probabilistic adjustment of beliefs about future life events can be considered as an approximation of a participant’s confidence in the future. The two free parameters of the Bayesian belief updating model scale how much the initial belief deviates from the updated, posterior belief (i.e., scaling parameter) and the propensity to consider the valence of this deviance (i.e., asymmetry parameter).

Although RL-like and Bayesian updating models make different assumptions about the updating strategy, they are complementary and powerful formalizations of human reasoning. Both models provide insight into hidden, latent variables of the updating process. Most notably, the learning rate and its components, the scaling and asymmetry parameters, which can vary between individuals and contexts and, through this variance, offer possible explanations for the idiosyncrasy in belief-updating behavior and its cognitive biases. “

Do the "winning" models produce the main behavioral patterns in Figure 1, and are they in some way uniquely able to do so, for instance? How would updating look different for an optimistic RL learner versus an optimistic Bayesian RL learner?

We now show that the winning models can reproduce the main behavioral patterns (revised Figure 1b).

Moreover, we plotted estimated and observed average belief updating for each participant (n=123) using the overall best-fitting asymmetrical RL-like updating model shown in SI Figure 6.

Would the asymmetry parameter in the former be correlated with the asymmetry parameter in the latter? Moreover, crucially, would one be able to reliably distinguish the models from one another under the model estimation and selection criteria that the authors have used here (presenting robust model recovery could help to show this)?

The asymmetry parameter estimated with the optimistically biased RL- and Bayesian models did correlate (r = 0.735; p < 0.001).

However, we argue that while the observed updating behavior and estimated free parameters are similar for RL-like and Bayesian learners, the underlying assumed cognitive processes differed and are identifiable. To test this assumption, we have added a model recovery analysis now reported in the supplement section 2c and main manuscript’s methods section pages 24–25.

As shown in SI Figure 5 confusion matrix, there is evidence for strong recovery of nearly all models, and importantly for the two winning models: the optimistically biased RL-like model and the rational Bayesian model of belief updating. This analysis thus rules out that the two model families were confused and mitigate concerns about the validity of the model selection.

Note, one exception was observed. The RL-like and Bayesian updating models that assumed no scaling and asymmetry were best recovered by their respective models that estimated the asymmetry parameter. Many factors could explain this. For example, it could be that the models, which assumed asymmetry, but no scaling, may have captured some bias in updating due to noise generated by the zero parameter models.

A justification is also needed to focus on the "RL-like updating model with an asymmetry and scaling learning rate component" in Figure 3. As I understand it, this model fits best outside of the pandemic, but another model - the Rational Bayesian Model - does worse (and does the best during the pandemic). What model best combines the groups (outside and inside the pandemic)?

We thank the reviewer for highlighting the need to justify our focus on the biased RL-like updating model in Figure 3. The model chosen for parameter comparison was selected based on a model comparison procedure conducted across all 12 models, including data from all participants (both those tested outside and during the pandemic, n=123). This model comparison revealed that Model 1 — the RL model with both asymmetry and scaling learning rate parameters estimated — provided the best fit across the entire dataset (Ef = 0.40, pxp = 0.99). As such, we focused on this model for parameter comparisons in Figure 3 to ensure consistency with the model comparison results and to interpret the parameters in the context of the overall best-fitting model. We added this information on top of the model parameter comparison results on page 8. Moreover, SI Figure 6 in the supplements shows how this model reproduces the observed belief updating in each of the 123 participants.

Why do the authors use absolute belief updating (|UPD|) in the first linear mixed effects model (equation iv)? Since an update is calculated differently depending on whether information calls for an update in an upward or downward direction, I do not understand the need to do this (and it means that updates that go in the wrong direction - away from the information - are counted as positive)

Thank you for driving our attention to this point. The ‘absolute belief updating’ note was incorrect, and we apologize for the confusion. To be precise, we did not use absolute updating values in our analyses. Belief updating was assumed on each trial to go either toward the base rate (e.g., Update = E2 – E1) for negative estimation errors or away from it for positive estimation errors (e.g., Update = E1 – E2). Updates that went in the wrong direction, further away from the base rate, were thus counted and included in the analysis with their negative sign. We have corrected this important point in equation iv of the methods section on page 19.

Figure 4: The task schema does not show a confidence rating for base rates.

Thank you for catching this. We have now added the confidence ratings for base rates to the task in Figure 4b in the revised version of the manuscript. We have furthermore corrected a typo in Figure 4a: The sample size for the group 3 tested in May 2021 now indicates 31.

The authors report that base rates are uniformly distributed - this is quite different to other instances of the task where base rates are normally distributed (ideally around the midpoint of the scale). Why this deviation in the design?

We used life events and base rates like those used in past studies of belief updating (Garrett and Sharot 2017, Sharot et al. 2011, Garrett et al. 2017, Korn et al. 2017), which were normal to uniformly distributed (W = 0.952, p = 0.088, Shapiro-Wilk test). The base rates ranged between 10% and 70%, with a mean of 40%. Participants rated their estimates between 3% and 77%, which ensured that for most likely (base rate = 70%) and most unlikely events (base rate = 10%) there was the same space (7%) to update beliefs toward the base rates. Moreover, all statistical models included the absolute estimation errors as a control for variance potentially explained by different estimation error magnitude[42,43]. We added this extra base rate information to the methods section’s task description on page 16.

The task is comprised of only negative life events, which arguably this hinders the generalizability of the results. The authors could mention this as a limitation (there has been a significant quantity of debate about this point in relation to this task: see the work from Ulrike Hahn's lab).

We have added a paragraph to the discussion page 13 to provide a rationale for using only adverse events. This paragraph now reads:

“In this study we tested how actual adverse experiences affect the updating of negative future outlooks in healthy participants and in analogy to studies conducted in depressed patients[19,20,24] following the cognitive model of depression[37]. One open question is whether findings were specific to the adverse event framing[38,39,40]. We argue that under normal, non-adverse contexts belief updating should also be optimistically biased for positive life events, as shown by previous research[41,42]. However, how context such as experiencing a challenging or favorable situation influence the updating of beliefs about positive and negative outlooks remains an open question.”

It would be useful to show the parameter recovery for all parameters (not just the learning rates) and the correlation between parameters (both in simulations and in the fitted parameters).

We apologize for being unclear on this part. The models included two free parameters that were the components of the learning rates: The scaling and the asymmetry parameter. We now have added parameter recovery analyses for the scaling and asymmetry components of the learning rates for (1) the Bayesian model of belief updating during the pandemic, and (2) the RL-like model of belief updating outside the pandemic to the supplement (SI section 2b, SI Figure 4).

**Reviewer #2:**
The authors investigated how experiencing the COVID-19 pandemic affected optimism bias in updating beliefs about the future. They ran a between-subjects design testing for participants on cognitive tasks before, during, and after lifting the sanitary state of emergence during the pandemic. The authors show that optimism bias varied depending on the context in which it was tested. Namely, it disappeared during COVID-19 and re-emerged at the time of lift of sanitary emergency measures. Through advanced computational modeling, they are able to thoroughly characterize the nature of such alternations, pinpointing specific mechanisms underlying the lack of optimistic bias during the pandemic.Strengths pertain to the comprehensive assessment of the results via computational modeling and from a theoretical point of view to the notion that environmental factors can affect cognition. However, the relatively small sample size for each group is a limitation.

Thank you for this review.

We acknowledge that sample sizes in each group are lower, especially when breaking down the participant sample into four sub-samples tested in the different contexts. To mitigate concerns we checked the power of the observed context by valence interaction on belief updating. To this aim we simulated new belief updates using the parameters from the best fitting optimistic RL-like model of observed belief updating outside the pandemic, and the rational Bayesian model of observed belief updating during the pandemic. At each iteration we performed a linear mixed effects model analysis of the simulated belief updates[44] analogous to equation iv in the main text. The frequency across 1000 iterations with which the LMEs detected a significant interaction of valence by context on simulated belief updating was 75 %. This frequency indicates the power of the valence by context interaction on observed belief updating. In other words, false negatives were 25% likely, which meant type II errors of failing to reject the null hypothesis when the effect was there. We have added these extra analyses to the main manuscript’s results section page 4 and method’s section page 20.

A major impediment interpreting of the findings is the need for additional measures. While the information on for example, risk perception or the need for social interaction was collected from participants during the pandemic, the fact that these could not be included in the analysis hinders the interpretation of findings, which is now generally based on data collected during the pandemic, for example, reporting increased stress. While authors suggest an interpretation in terms of uncertainty of real-life conditions it is currently difficult to know if that factor drove the effect. Many concurrent elements might have accounted for the findings. This limits understanding of the underlying mechanisms related to changes in optimism bias.

We agree with the reviewer on the limitation arising from the lack of physiological and self-report measures of stress, threat, and perceived uncertainty. To address this point and a similar point raised by reviewer 1 we have added a section to the supplement (SI section 4) that now reports explorative correlations between questionnaire responses of subjective perceptions of risk and anxiety, behavior (e.g. mask wearing, social distancing) and belief updating measured during the 1st strict lockdown.

We now also further discuss this limitation on page 12 of the main text’s discussion.

I recommend that the authors spend more time on explaining the belief-updating task in the presentation of the experiment.

Thank you for this advice. We now provide a clearer and more detailed description of the belief-updating task in the main manuscript’s methods section and have updated Figure 4b to display the confidence rating event in the task schema.

The task description now reads:

“As illustrated in Figure 4b, each of the 40 trials began with presenting an adverse life event. Participants estimated their own risk and the risk of someone else their age and gender. Then the base rate of the event occurring in the general population was displayed on the computer screen. Participants rated their confidence in the accuracy of the presented base rate. Finally, they re-estimated their risk for experiencing the event now informed by the base rate.”

The experimental task seems to include a self-other dimension, which is completely disregarded in the analysis. It would be interesting to explore whether the effect of diluted optimism bias during the pandemic is specific to information about self vs. Other.

We appreciate the reviewer's observation regarding the self-versus-other dimension in the belief updating task design. As now shown in SI Figure 2 the participants indeed displayed an optimism bias: They estimated that adverse events are more likely to happen to others than to themselves (ß = 3.02, SE = 0.86, t (232) = 3.53, p = 5.09e-04, 95% CI [1.33 – 4.71]; SI Figure 2; SI Table 18). This effect was observed overall participants. The pandemic context had no significant effect (ß = -1.91, SE = 3.00, t (232) = -0.64, p = 0.52, 95% CI [-7.82 – 4.00]; SI Table 18). Moreover, following previous studies of optimistically biased belief updating we tested the effect of estimation errors (EE) calculated on the difference between the estimate for someone else (eBR) and the base rate (BR), following: EE = eBR – BR[4,5,25,26]. When categorizing trials as good news or bad news based on this alternative EE calculation the context-by-EE valence interaction remained significant (SI Table 6).

We conclude from these additional analyses that experiencing the pandemic specifically influenced belief updating but did not affect optimism biases in initial beliefs about the future.

Please provide an English translation of the instructions for the task.

We now provide an English translation of the task instructions in the Supplement section 5.